# Physiologically relevant acid-sensing ion channel (ASIC) 2a/3 heteromers have a 1:2 stoichiometry

Leon Fischer[1,3,6], Axel Schmidt[1,4,6], Anke Dopychai[2], Sylvia Joussen[1], Niko Joeres[1,5], Adrienne Oslender-Bujotzek[1], Günther Schmalzing[2] & Stefan Gründer [1✉]

Acid-sensing ion channels (ASICs) sense extracellular protons and are involved in synaptic transmission and pain sensation. ASIC1a and ASIC3 are the ASIC subunits with the highest proton sensitivity. ASIC2a in contrast has low proton sensitivity but increases the variability of ASICs by forming heteromers with ASIC1a or ASIC3. ASICs are trimers and for the ASIC1a/2a heteromer it has been shown that subunits randomly assemble with a flexible 1:2/2:1 stoichiometry. Both heteromers have almost identical proton sensitivity intermediate between ASIC1a and ASIC2a. Here, we investigated the stoichiometry of the ASIC2a/3 heteromer. Using electrophysiology, we extensively characterized, first, cells expressing ASIC2a and ASIC3 at different ratios, second, concatemeric channels with a fixed subunit stoichiometry, and, third, channels containing loss-of-functions mutations in specific subunits. Our results conclusively show that only ASIC2a/3 heteromers with a 1:2 stoichiometry had a proton-sensitivity intermediate between ASIC2a and ASIC3. In contrast, the proton sensitivity of ASIC2a/3 heteromers with a 2:1 stoichiometry was strongly acid-shifted by more than one pH unit, which suggests that they are not physiologically relevant. Together, our results reveal that the proton sensitivity of the two ASIC2a/3 heteromers is clearly different and that ASIC3 and ASIC1a make remarkably different contributions to heteromers with ASIC2a.

[1] Institute of Physiology, RWTH Aachen University, Pauwelsstrasse 30, D-52074 Aachen, Germany. [2] Institute of Clinical Pharmacology, RWTH Aachen University, Wendlingweg, D-52074 Aachen, Germany. [3] Present address: Department of Anesthesiology, Technical University Dresden, Dresden, Germany. [4] Present address: Institute of Human Genetics, University of Bonn, School of Medicine & University Hospital Bonn, Bonn, Germany. [5] Present address: Department of Nephrology and Clinical Immunology, RWTH Aachen University, Pauwelsstrasse 30, D-52074 Aachen, Germany. [6] These authors contributed equally: Leon Fischer, Axel Schmidt. ✉email: sgruender@ukaachen.de

Acid-sensing ion channels (ASICs) are proton-gated $Na^+$ channels widely expressed in neurons[1]. In the central nervous system (CNS), they have a dendritic localization and contribute to the excitatory postsynaptic current[2,3]. In the peripheral nervous system (PNS), they sense painful acidosis[4]. Moreover, they contribute to the pathophysiological outcome of several neuronal disorders[5], such as stroke and multiple sclerosis[6,7], rendering ASICs interesting drug targets.

There are six ASIC subunits, ASIC1a, ASIC1b, ASIC2a, ASIC2b, ASIC3, and ASIC4[8], which assemble into trimeric channels[9,10]. Functional ASICs can be either homotrimers or heterotrimers. While homomeric ASIC1a and ASIC3 have a high proton sensitivity with a pH of half-maximal activation ($pH_{50}$) of ~6.5[11-13], homomeric ASIC1b and in particular ASIC2a have lower proton sensitivity ($pH_{50}$ of ASIC1b ~ 6.0 and $pH_{50}$ of ASIC2a ~ 4.0)[13-15]; homomeric ASIC2b and ASIC4 are insensitive to protons[16-18].

In the CNS, the main ASICs are homomeric ASIC1a and heteromeric ASIC1a/2a and ASIC1a/2b[19-24]. It has been shown that the ASIC1a/2a heteromer has a flexible stoichiometry of 1:2 and 2:1[10]. Strikingly, both ASIC1a/2a heteromers have very similar functional properties, in particular $pH_{50}$[10,25], suggesting that individual subunits do not additively contribute to the functional properties of an ASIC, such as $pH_{50}$. Moreover, toxins and other modulators of ASICs bind at subunit interfaces[26,27], rendering the knowledge of subunit stoichiometry of interest for drug development. Yet, apart from the ASIC1a/2a heteromer, for no other ASIC heteromer has the subunit stoichiometry been firmly established. In the PNS, the ASIC2a/3 heteromer is important: it is expressed in mouse dorsal root ganglion neurons, in particular in cardiac afferents[28,29], where it senses acidosis during myocardial ischemia and contributes to chest pain. Moreover, it allows a direct comparison of the influence of ASIC3 with that of ASIC1a on a heteromeric ASIC.

In this study, we characterized the stoichiometry of the ASIC2a/3 heteromer by different molecular, biochemical, and electrophysiological approaches. In contrast to the ASIC1a/2a heteromer, we found striking functional differences between the ASIC2a/3 heteromer with a 1:2 and a 2:1 stoichiometry. While the ASIC2a/3 heteromer with a 1:2 stoichiometry had characteristics that are intermediate between ASIC2a and ASIC3, the ASIC2a/3 heteromer with a 2:1 stoichiometry had characteristics that resembled homomeric ASIC2a. Since the pH values required to activate the ASIC2a/3 heteromer with 1:2 stoichiometry are much more in the physiological range, our data suggest that this heteromer is the physiologically relevant one.

## Results

### Functional characterization of ASICs in oocytes co-expressing ASIC2a and ASIC3 at variable ratios.
We characterized ASIC2a, ASIC3, and their putative heteromers by two-electrode voltage clamp in *Xenopus laevis* oocytes. In oocytes expressing only ASIC3, acidic solutions elicited currents that desensitized in <1 s; at pH 5.5 or lower a non-desensitizing current component appeared that had an amplitude of ~2% of the peak amplitude. Apparent $pH_{50}$ of the transient peak current was $6.40 \pm 0.01$ and the Hill-coefficient was $2.5 \pm 0.1$ (Fig. 1a, c, Table 1). Changing the conditioning pH from pH 7.4 to slightly more acidic values revealed half-maximal steady-state desensitization (SSD) at pH $7.21 \pm 0.02$ (Hill-coefficient $-13.0 \pm 1.8$; Fig. 1b, e, Table 1). In contrast, in oocytes expressing only ASIC2a, acidic solutions elicited currents that did not fully desensitize within the 10 s of acid application and apparent $pH_{50}$ was acid-shifted by more than two pH units compared with ASIC3 ($pH_{50}$ $3.96 \pm 0.05$, Hill-coefficient $0.8 \pm 0.03$; Fig. 1a, c, Table 1); half-maximal SSD was

reached at pH $6.5 \pm 0.06$ (Hill-coefficient $-2.1 \pm 0.2$; Fig. 1b, e, Table 1). Considering that the precision with which the $pH_{50}$ of activation of ASIC2a can be determined is limited by the use of very acidic solutions, these values agree well with previously published values[12,30-32].

To characterize the functional properties of putative ASIC2a/3 heteromers, we injected oocytes with varying ratios of ASIC2a and ASIC3 cRNA (5:1, 1:1, 1:5, and 1:10). With increasing amounts of ASIC3 cRNA, apparent $pH_{50}$ of ASICs in these oocytes was alkaline-shifted from pH $3.43 \pm 0.18$ (ASIC2a:ASIC3 5:1) to pH $5.69 \pm 0.05$ (1:10; $p < 10^{-7}$; ANOVA followed by Tukey's test; Fig. 1a, d, g, Table 1). Half-maximal SSD was also alkaline-shifted from pH $6.36 \pm 0.01$ (5:1) to pH $6.77 \pm 0.01$ (1:10; $p < 10^{-7}$; ANOVA followed by Tukey's test; Fig. 1b, f, g, Table 1). These shifts in the values for $pH_{50}$ of activation and of SSD indicate a mixed ASIC population after injection of different ASIC2a:ASIC3 cRNA ratios.

To determine the contribution of ASIC2a/3 heteromers to these mixed ASIC populations, we fitted the concentration-response relationships with the sum of three Hill-functions (Fig. 2). For two of these Hill functions, we used $pH_{50}$ and Hill-coefficients as determined above for homomeric ASIC2a and homomeric ASIC3, respectively. The third Hill-function was not restrained and therefore could capture the properties of potential ASIC heteromers. We determined the relative contributions of the three Hill-functions to the total current as well as the $pH_{50}$ and Hill-coefficient of the third Hill-function (see "Methods"). The contribution of the third Hill function (representing the heteromeric population) to the total current ranged from 16 to 86% and was consistently high (62 to 86%) for the recordings with ASIC2a:ASIC3 ratios of 1:5 and 1:10 (Table 2). The $pH_{50}$ values of the heteromeric population in these experiments were $5.51 \pm 0.03$ and $5.57 \pm 0.03$ for activation and $6.88 \pm 0.13$ and $6.76 \pm 0.01$ for SSD (Fig. 2 and Table 2), respectively, suggesting that heteromeric ASIC2a/3 channels have a proton-sensitivity in-between the two homomers.

### ASIC2a-3 concatemers form functional trimers in oocytes.
Because curve fitting assuming a scenario with two different heteromers (with a 1:2 and a 2:1 stoichiometry) co-existing with the two homomers cannot faithfully discriminate from a scenario with one heteromer co-existing with two homomers[10], we turned to concatemers, in which we fixed the stoichiometry to either 1:2 or 2:1 by covalently linking subunits in the desired configurations. Previously, we could successfully characterize the two individual ASIC1a/2a heteromers by constructing such concatemeric channels with the desired subunit composition[25]. Therefore, we generated similar trimeric concatemers containing different numbers of ASIC2a and ASIC3 subunits in different orders. For detection of the concatemers in western blots, we also constructed concatemers with an N-terminal heptahistidine-tag (His-tag; see "Methods" section). These His-tagged constructs had similar properties as untagged concatemers (see below; Fig. 3c, d).

We then biochemically examined the synthesis and plasma membrane appearance of full-length ASIC2a/3 concatemers compared with individually expressed (non-concatenated) ASIC2a and ASIC3 subunits. The respective constructs were dyed for detection, purified via the N-terminal His-tag, resolved by SDS-urea-PAGE in non-reduced and DTT-reduced states, and visualized by infrared and radio scanning. ASIC2a migrated at the expected calculated total mass of ~64 kDa (59 kDa protein core plus ~5 kDa provided by two N-glycans at $^{365}$NLT and $^{392}$NKS) in both $^{35}$S-labeled form and plasma membrane form (Supplementary Fig. 1, lane 1). A faster migrating band of ~50 kDa is probably a degradation product of ASIC2a (lane 1). Its

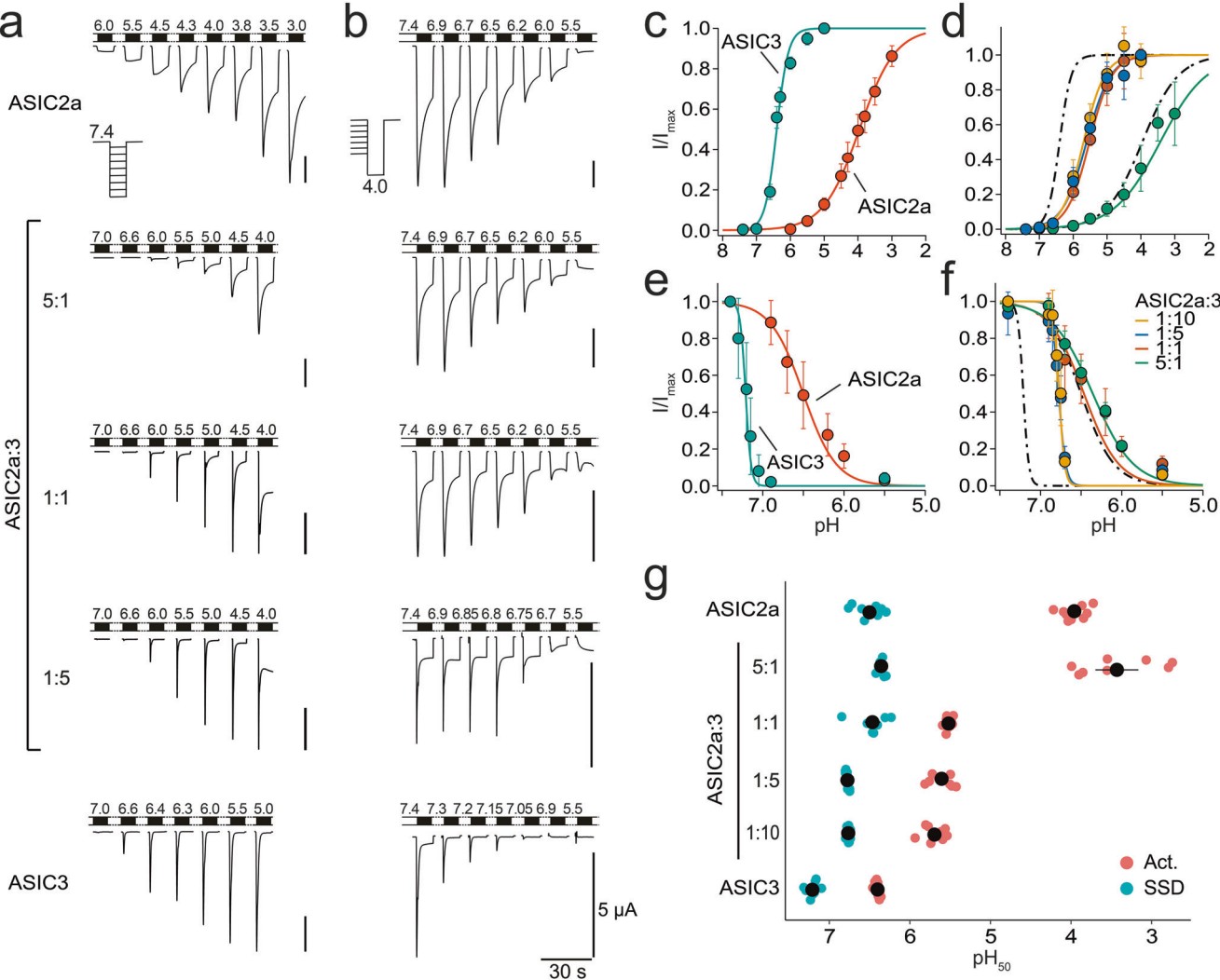

**Fig. 1 pH response relationships for ASIC2a, ASIC3, and oocytes co-expressing ASIC2a and ASIC3. a**, **b** Representative current traces from measurements to determine $pH_{50}$ of activation (**a**) and of SSD (**b**). For SSD measurements, activation pH was 4.0. **c**–**f** Relative mean current amplitudes of oocytes expressing ASIC2a, ASIC3 (**c**, **e**) or co-expressing ASIC2a and ASIC3 at different cRNA ratios (**d**, **f**) at varying pH values. The pH values indicated were used for activation (**c**, **d**) or pre-conditioning (**e**, **f**) of oocytes. Solid lines represent fits to a Hill function. The dotted lines in panels d and f represent fits of the homomers from panels (**c**) and (**e**). ASIC2a and ASIC2a/3 5:1 did not reach a plateau in the pH range under investigation and corresponding Hill functions were therefore fitted with open constrains and rescaled afterwards. All other Hill functions were fixed to a maximum current amplitude of 1 (see "Methods" for details). **g** Overview of individual and average $pH_{50}$ values (colored and black points, respectively) for activation (Act.) and SSD. Error bars represent SEM. For the number n of independent oocytes and for absolute values of $I_{max}$, see Table 1.

**Table 1 Electrophysiological characteristics of ASIC2a and ASIC3 homomers and of ASICs from oocytes co-expressing ASIC2a and ASIC3 at different cRNA ratios.**

| | 2a:3 ratio | Act | | | SSD | | | $I_{max}$ (µA) | $I_{5s}/I_{peak}$ (%) | $\tau_{des}$ (s) |
|---|---|---|---|---|---|---|---|---|---|---|
| | | $pH_{50}$ | Hill | n | $pH_{50}$ | Hill (SSD) | n | | | |
| ASIC2a | – | 3.96 ± 0.05 | 0.8 ± 0.0 | 9 | 6.50 ± 0.06 | −2.1 ± 0.2 | 9 | −18.7 ± 2.6 | 76 ± 4 | 2.1 ± 0.1 |
| mixture | 5:1 | 3.43 ± 0.18 | 0.6 ± 0.1 | 8 | 6.36 ± 0.01 | −1.7 ± 0.1 | 8 | −13.5 ± 1.2 | 43 ± 4 | 1.5 ± 0.1 |
| | 1:1 | 5.54 ± 0.01 | 1.4 ± 0.1 | 9 | 6.47 ± 0.06 | −2.0 ± 0.2 | 8 | −9.0 ± 1.2 | 12 ± 1 | 1.1 ± 0.1 |
| | 1:5 | 5.61 ± 0.05 | 1.2 ± 0.1 | 8 | 6.78 ± 0.01 | −10.1 ± 0.6 | 10 | −6.5 ± 0.8 | 8 ± 1 | 0.7 ± 0.1 |
| | 1:10 | 5.69 ± 0.05 | 1.4 ± 0.1 | 8 | 6.77 ± 0.01 | −13.6 ± 1.9 | 10 | −6.0 ± 0.7 | 3 ± 0 | 0.9 ± 0.1 |
| ASIC3 | – | 6.40 ± 0.01 | 2.5 ± 0.1 | 8 | 7.21 ± 0.02 | −13.0 ± 1.8 | 9 | −9.4 ± 1.8 | 2 ± 0 | 0.5 ± 0.0 |

$pH_{50}$ values and Hill coefficients are given for activation (Act) and SSD. n = number of independent oocytes; $I_{max}$ = mean maximum current amplitude; $I_{5s}/I_{peak}$ = ratio of the current amplitude 5 s after activation and the peak current amplitude at pH 4.5; $\tau_{des}$ = time constant of desensitization at pH 4.0. Values indicate mean ± SEM.

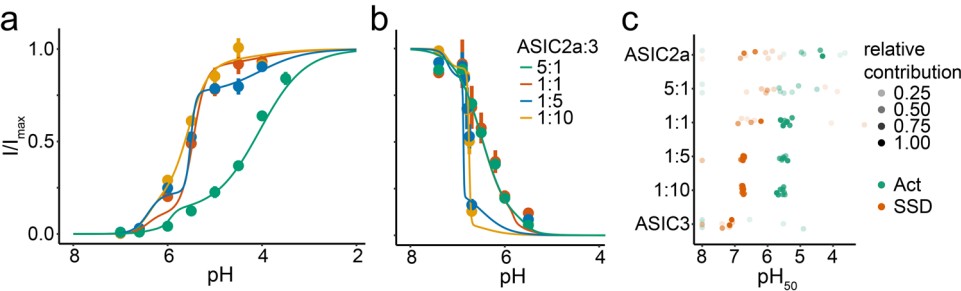

**Fig. 2 Analysis of mixed ASIC populations. a, b** pH response relationships for activation (**a**) and SSD (**b**) from Fig. 1 were reanalyzed by assuming a mixed ASIC population and by fitting with the sum of three Hill functions. Two Hill functions had fixed $pH_{50}$ values and Hill coefficients corresponding to ASIC2a and ASIC3, respectively, and the third Hill function was flexible. Averaged fits are shown as solid lines, error bars represent the SEM. For better comparison, the fits were scaled to a range between 0 and 1 and relative currents were scaled accordingly. **c** $pH_{50}$ values of the third, flexible Hill function are indicated for individual recordings, $pH_{50}$ values of activation in bluish green and $pH_{50}$ of SSD in brownish. Tints illustrate the contribution of the heteromeric population to the total population. For the number n of independent oocytes, see Table 2.

**Table 2 Fit of mixed ASIC populations by the sum of three Hill functions.**

| 2a:3 ratio | Protocol | Contribution | $pH_{50}$ | Hill | n |
|---|---|---|---|---|---|
| 5:1 | Act | 0.20 ± 0.04 | 5.94 ± 0.47 | 5.1 ± 2.6 | 8 |
|  | SSD | 0.16 ± 0.05 | 5.54 ± 0.40 | −4.5 ± 2.5 | 7 |
| 1:1 | Act | 0.83 ± 0.04 | 5.46 ± 0.04 | 3.3 ± 1.7 | 9 |
|  | SSD | 0.27 ± 0.12 | 5.82 ± 0.51 | −14.4 ± 5.4 | 8 |
| 1:5 | Act | 0.62 ± 0.07 | 5.51 ± 0.03 | 8.2 ± 2.9 | 8 |
|  | SSD | 0.73 ± 0.06 | 6.88 ± 0.13 | −53.7 ± 14.1 | 10 |
| 1:10 | Act | 0.77 ± 0.04 | 5.57 ± 0.03 | 2.0 ± 0.4 | 8 |
|  | SSD | 0.86 ± 0.04 | 6.76 ± 0.01 | −25.3 ± 9.4 | 9 |

For the Hill function representing the heteromer, its contribution to the total current and the $pH_{50}$ value as well as the Hill-coefficients are indicated. Values indicate mean ± SEM. n = number of independent oocytes.

proportion is overestimated due to the comparably high amplification of fluorescence; quantitatively, it accounts for only 5% of ASIC2a. A 120 kDa band is also visible in the plasma membrane in lane 1, which is most likely a ASIC2a dimer. In contrast to ASIC2a, ASIC3 (60 kDa protein core plus ~5 kDa provided by two N-glycans at [176]NFT and [400]NRS) was only visible in the [35]S labeled form, but not at the plasma membrane (lane 18), which we attribute to the overall lower expression of the ASIC3 protein.

All the ASIC2a-ASIC3 concatemers were visible in both the [35]S-labeled and plasma membrane-bound forms, and predominantly in masses consistent with full-length ASIC2a/3 trimers (lanes 2–15). The slower migration of the concatemers in the DTT-reduced state is consistent with the reduction of intrasubunit disulfide bonds present in the ASICs ectodomains. The breaking of these disulfide bonds relaxes the compactness of the protein structure, thus resulting in an expected slower migration in the SDS-urea-PAGE gel. Small amounts of lower order byproducts consisting of monomers and dimers (as indicated by the number of dots in the bands) are also seen in both the [35]S-labeled and plasma membrane-bound forms, but only when the ASIC2a-ASIC3 concatemers contained two copies of ASIC2a (lanes 2–9). In contrast, no such byproducts were observed when the concatemers contained two copies of ASIC3 (lanes 10–15). The concatenated homotrimer ASIC3 (3-3-3) was only very weakly expressed in the [35]S labeled form and was not detected at the plasma membrane (lanes 16–17), like monomeric ASIC3 (lane 18).

**The two ASIC2a/3 heteromers have widely different proton sensitivity.** Next, we functionally characterized the concatemers and compared them to ASIC channels assembled from individual (non-concatenated) subunits. We observed proton-sensitive currents for all constructs, also the homotrimer 3-3-3 (Fig. 3a–c) and determined their pH dependence of activation (see "Methods"). First, we characterized concatemers containing just ASIC2a subunits. Although the $pH_{50}$ of the 2a-2a-2a concatemer (pH 4.10 ± 0.16) was significantly different from ASIC2a (pH 3.38 ± 0.02; $p < 0.001$; ANOVA followed by Tukey's test; Fig. 3d; Table 3), we note that the acidic pH necessary to activate ASIC2a renders the determination of $pH_{50}$ values less precise. Replacing one ASIC2a subunit by an ASIC3 subunit in the concatemers yielded channels with $pH_{50}$ values in the range of 3.93-4.14 ($p = 0.99$ for all comparisons within this group; Fig. 3d, e, Table 3), like the 2a-2a-2a concatemer ($p = 0.94$–0.99). In contrast, replacing two ASIC2a subunits by ASIC3 subunits strongly shifted $pH_{50}$ to the range of 5.29-5.65 ($p = 0.11$–0.99 for all comparisons within this group) which was significantly different ($p < 0.001$) from ASIC2a, the 2a-2a-2a concatemer and the concatemers containing only one ASIC3 subunit (Fig. 3d, e, Table 3). Finally, the 3-3-3 concatemer had a $pH_{50}$ like ASIC3 (pH 6.39; Fig. 3d, Table 3), which was significantly different from ASIC2a and all the other concatemers ($p < 0.001$).

We determined the pH dependence of SSD for ASIC2a and ASIC3 as well as for four representative concatemers (2a-2a-2a, 3-3-3, and the heteromers 2a-3-2a and 3-2a-3). ASIC2a and the 2a-2a-2a concatemer had $pH_{50}$ values of 6.24 ± 0.03 and 6.03 ± 0.09, respectively ($p = 0.009$; ANOVA followed by Tukey's test; Fig. 3d, Table 3). Like the $pH_{50}$ of activation, the $pH_{50}$ of SSD of the heteromeric concatemer 2a-3-2a, containing one ASIC3 subunit, was similar to the $pH_{50}$ of ASIC2a (6.19 ± 0.02; $p = 0.92$). In contrast, the heteromeric concatemer 3-2a-3, containing two ASIC3 subunits, had a $pH_{50}$ of 6.59 ± 0.01, which was significantly higher than for ASIC2a and the concatemers 2a-2a-2a and 2a-3-2a ($p < 0.001$; Fig. 3d, e, Table 3). ASIC3 and the 3-3-3 concatemer had similar $pH_{50}$ values (7.14 ± 0.02 and 7.17 ± 0.01, respectively; $p = 0.99$; Fig. 3d), which were significantly different from all other constructs ($p < 0.001$).

For the SSD measurements, we also determined time constants of desensitization ($\tau_{des}$) at pH 4.0 with an automated curve-fitting procedure (Table 3). ASIC2a had a $\tau_{des}$ of 10.8 ± 3.1 s ($n = 4$) and the 2a-2a-2a concatemer had a similar $\tau_{des}$ of 9.9 ± 2.3 s ($n = 8$; $p = 0.99$, ANOVA followed by Tukey's test). Interestingly, all other channels had relatively similar $\tau_{des} < 2$ s ($\tau_{des}$ of 2a-3-2a: 1.7 ± 0.4 s, $n = 7$; $\tau_{des}$ of 3-2a-3: 0.6 ± 0.05 s, $n = 9$; $\tau_{des}$ of ASIC3: 0.6 ± 0.05 s, $n = 9$; $\tau_{des}$ of 3-3-3: 0. 6 ± 0.03 s, $n = 10$), which were

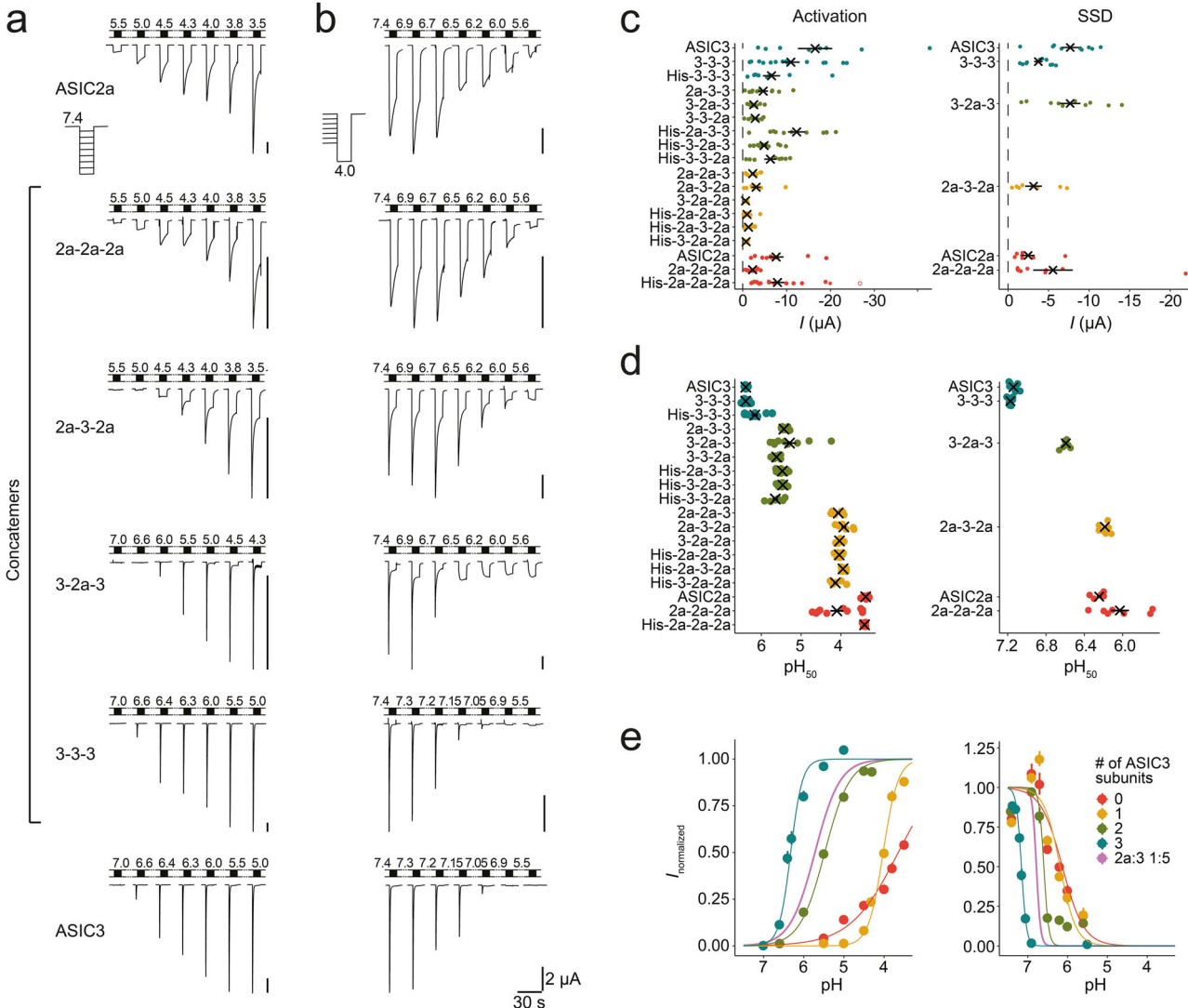

**Fig. 3 Functional characterization of ASIC2a-ASIC3 concatemers. a, b** Representative current traces from measurements to determine pH$_{50}$ of activation (**a**) and of SSD (**b**) for ASIC2a-ASIC3 concatemers. **c** Current amplitudes of individual oocytes (dots) and mean current amplitudes (crosses) are shown, the color code indicates the number of ASIC3 subunits. For activation recordings, the current amplitudes at pH 3.5 (ASIC2a or concatemers containing no or one ASIC3 subunit), pH 4.3 (concatemers containing two ASIC3 subunits), or pH 5 (ASIC3 or 3a-3a-3a) are shown. For SSD recordings, current amplitudes at pH 4.0 are shown. **d** pH$_{50}$ of activation and of SSD were determined by fitting peak currents with Hill functions. Values for individual oocytes are shown as dots, mean values as squares. **e** pH response curves for ASICs containing zero to three ASIC3 subunits per trimer are shown. For activation curves, current values as well as Hill function parameters were averaged for all ASICs with the respective number of ASIC3 subunits, including His-tagged constructs. For comparison, the activation and SSD curves of oocytes expressing ASIC2a and ASIC3 at a 1:5 ratio from Fig. 1 are shown. Error bars represent SEM. For the number n of independent oocytes, see Table 3.

**Table 3 Electrophysiological characteristics of ASIC2a-ASIC3 concatemers.**

|  | Act | | | SSD | | | $I_{pH4}$ (μA) | $I_{5s}/I_{peak}$ (%) | $\tau_{des}$ (s) |
|---|---|---|---|---|---|---|---|---|---|
|  | pH$_{50}$ | Hill | n | pH$_{50}$ | Hill | n |  |  |  |
| ASIC3 | 6.39 ± 0.01 | 3.1 ± 0.3 | 7 | 7.14 ± 0.02 | −6.1 ± 0.5 | 8 | −7.7 ± 1.0 | 5 ± 2 | 0.6 ± 0.1 |
| 3-3-3 | 6.39 ± 0.02 | 3.1 ± 0.1 | 14 | 7.17 ± 0.01 | −7.4 ± 0.5 | 9 | −3.7 ± 0.5 | 3 ± 1 | 0.6 ± 0.0 |
| 3-2a-3 | 5.29 ± 0.17 | 1.4 ± 0.2 | 9 | 6.59 ± 0.01 | −7.1 ± 1.1 | 8 | −7.7 ± 1.3 | 13 ± 1 | 0.6 ± 0.1 |
| 2a-3-2a | 3.93 ± 0.06 | 1.9 ± 0.2 | 8 | 6.19 ± 0.02 | −2.2 ± 0.2 | 7 | −3.2 ± 1.0 | 42 ± 7 | 1.9 ± 0.4 |
| 2a-2a-2a | 3.66 ± 0.05 | 0.7 ± 0.1 | 10 | 6.03 ± 0.09 | −1.4 ± 0.2 | 7 | −5.5 ± 2.4 | 71 ± 4 | 9.9 ± 2.3 |
| ASIC2a | 3.38 ± 0.02 | 0.6 ± 0.0 | 8 | 6.24 ± 0.03 | −1.9 ± 0.2 | 6 | −2.5 ± 0.8 | 75 ± 6 | 10.8 ± 3.1 |

pH$_{50}$ values and Hill coefficients are given for activation (Act) and SSD. n = number of independent oocytes; $I_{pH4}$ = current amplitude when activated with pH 4.0; $I_{5s}/I_{peak}$ = ratio of the current amplitude 5 s after activation and the peak current amplitude at pH 4.0; $\tau_{des}$ = time constant of desensitization at pH 4.0. Values indicate mean ± SEM.

not different from each other ($p = 0.96-0.99$) but significantly different from ASIC2a and 2a-2a-2a ($p < 0.001$). It should be noted that the determination of $\tau_{des} < 1$ s is limited in oocytes by the comparatively slow solution exchange. Nevertheless, in contrast to $pH_{50}$ values, $\tau_{des}$ allowed to differentiate the concatemer containing one ASIC3 subunit from ASIC2a and the 2a-2a-2a concatemer. Similarly, the current remaining 5 s after the peak current ($I_{5s}$) was small for homomeric ASIC3 ($5 \pm 2\%$ of the peak), slightly larger for the 3-2a-3 concatemer ($13 \pm 1\%$; $p = 0.52$ compared to ASIC3, ANOVA followed by Tukey's test), large for homomeric ASIC2a (>70%; $p < 0.001$ when compared to any other ion channel composition), and intermediate for the 2a-3-2a concatemer ($42 \pm 7\%$; $p < 0.001$ when compared to any other ion channel composition; Table 3).

Collectively, these results reveal that concatemers containing two ASIC3 and one ASIC2a subunit form an ASIC population with properties clearly different from the two homomeric populations. In contrast, results for concatemers containing one ASIC3 and two ASIC2a subunits were ambiguous. These channels had properties like ASIC2a, and activation required strong acidification to pH values below pH 5.0. Only $\tau_{des}$ was different between ASIC2a homotrimers and concatemers containing one ASIC3 and two ASIC2a subunits. In any case, the two heteromers clearly had different proton sensitivities, which is in sharp contrast to the two ASIC1a/2a heteromers[25]. Intriguingly, $pH_{50}$ values, $\tau_{des}$, and $I_{5s}/I_{peak}$ for concatemers containing two ASIC3 and one ASIC2a subunit closely resembled the properties of the putative heteromeric population determined in oocytes expressing ASIC2a together with ASIC3 (Fig. 2; Table 2).

**Mutation of a critical His residue in individual subunits reports the number of mutant subunits.** To further corroborate our results, we turned to the analysis of trimers containing varying numbers of a loss-of-function mutation. Combined substitution of the histidine residues at positions 72 and 73 in ASIC1a by asparagine (H72N_H73N) leads to complete loss-of-function[33]. We reasoned that if the loss-of-function depends on the number of subunits with the His-to-Asn exchange, this mutation might be useful to report the number of a specific subunit in a channel. We tested this assumption using an ASIC1a concatemer in which three ASIC1a subunits were covalently linked (1a-1a-1a). This concatemer expressed well in oocytes (Fig. 4a) and displayed electrophysiological properties that resembled ASIC1a wild type closely (see Supplementary Table 1).

We then generated all possible ASIC1a concatemers with one to three subunits carrying the H72N_H73N exchange and studied them by electrophysiology (Fig. 4b; Supplementary Table 1). We injected equal RNA amounts of ASIC1a or the different ASIC1a concatemers and determined the current amplitude at pH 4.0. Oocytes expressing ASIC1a had an average current amplitude of $-32.71 \pm 3.87$ µA ($n = 9$). The 1a-1a-1a concatemer had approximately threefold smaller current amplitudes of $-11.70 \pm 1.85$ µA ($n = 7$; $p < 10^{-7}$; ANOVA followed by Tukey's test; Fig. 4c). Compared with the wild-type 1a-1a-1a concatemer, concatemers with one mutated subunit had more than tenfold smaller current amplitudes, irrespective of the position of the mutated subunit within the channel ($-0.89 \pm 0.02$ µA, $n = 22$; $p < 10^{-7}$; ANOVA followed by Tukey's test; Fig. 4c). As expected, mutation of all three subunits led to a loss-of-function (current amplitude $-0.03 \pm 0.005$ µA, $n = 6$). Strikingly, when comparing mutated concatemers, mutation of two subunits was sufficient to lead to a loss-of-function and current amplitudes were indeed not significantly different from current amplitudes of triple-mutant concatemers ($-0.14 \pm 0.01$ µA, $n = 23$; $p = 0.802$; Fig. 4c). In contrast, concatemers with one mutant subunit had significantly

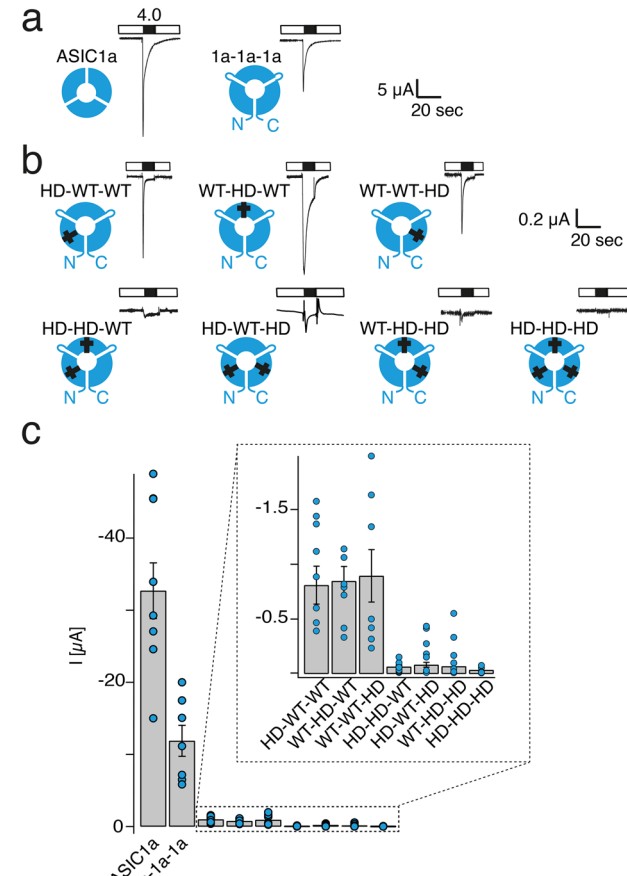

**Fig. 4 Mutation of two ASIC subunits leads to a loss-of-function.**
**a** Representative current traces of ASIC1a and the 1a-1a-1a concatemer. **b** Representative current traces of concatemers carrying a H72N_H73N mutation in different subunits; mutated subunits are marked by a cross. **c** Mean current amplitudes for activation with pH 4.0 for 10 s. Oocytes had been injected with equal amounts of cRNA (~0.5 ng). Error bars represent SEM. The number $n$ of independent oocytes was (from left to right): 9, 7, 8, 7, 7, 8, 11, 7, and 6, respectively.

higher current amplitudes than concatemers with three or two mutant subunits ($p(1vs3) < 10^{-5}$; $p(1vs2) < 10^{-7}$; ANOVA followed by Tukey's test).

We next measured concentration-response curves for all concatemers in the range pH 7.4−4.0. The amplitudes of resulting currents were fitted by a Hill function. Concatemers with one mutant subunit had a $pH_{50}$ of $6.51 \pm 0.02$ when the first subunit contained the mutation (HN-WT-WT), $6.08 \pm 0.05$ when the second subunit contained the mutation (WT-HN-WT; vs. HN-WT-WT: $p < 10^{-7}$, ANOVA followed by Tukey's test) and $5.96 \pm 0.05$ when the third subunit contained the mutation (WT-WT-HN, vs. HN-WT-WT: $p < 10^{-7}$, vs. WT-HN-WT: $p = 0.121$, ANOVA followed by Tukey's test) (Supplementary Table 1). It is interesting to note that the concatemer with a mutated subunit in the first position had a proton sensitivity that was significantly different from the other two concatemers. We currently have no explanation for this position-dependence of the mutation. The average $pH_{50}$ for concatemers with one mutant subunit was $6.24 \pm 0.04$ ($n = 55$; vs. WT concatemer: $p = 0.012$, ANOVA followed by Tukey's test), indicating a slight acid-shift by one mutant subunit. Consistent with a loss-of-function of ASIC1a concatemers with two or three mutant subunits, reliable fitting of peak currents by the Hill function was not possible.

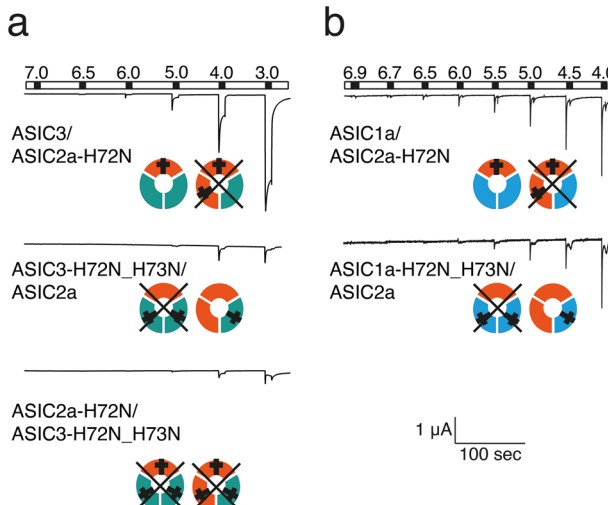

**Fig. 5 Only ASIC3 but not ASIC2 can rescue a heteromer containing mutant subunits. a** Representative current traces of ASIC2a-H72N with ASIC3-wt and of ASIC2a-wt with ASIC3-H72N_H73N. The two possible heteromers are indicated below the traces (ASIC2a in red, ASIC3 in green); mutated subunits are marked by a cross. Channels with two mutant subunits are crossed out, assuming that they are inactive. Note that only when ASIC3-wt ($n = 9$) but not when ASIC2a-wt ($n = 8$) was present, robust ASIC activity could be measured, suggesting that the channel with two ASIC2a subunits made no contribution to the currents. Similarly, co-injection of mutated ASIC2a with mutated ASIC3 showed minimal activity and only at strongly acidic pH ($n = 8$). **b** As in (**a**), but with ASIC1a instead of ASIC3. Here, robust ASIC activity could be measured also when ASIC2a-wt was present. Error bars represent SEM. $n$ = number of independent oocytes.

**Functional characterization of ASIC2a/3 heteromers containing individual mutant subunits**. To investigate the stoichiometry of ASIC2a/3 heteromers, we injected ASIC2a-wt or ASIC2a-H73N together with ASIC3-wt or ASIC3-H72N_H73N into oocytes and compared current amplitudes and $pH_{50}$ of the resulting ASICs. We injected ratios of cRNAs that should lead to predominantly heteromeric channels (see Table 2). Channels were activated with pH ranging from 7.0 to 3. In oocytes expressing only mutant subunits (ASIC2a-H72N and ASIC3-H72N_H73N) only pH 4.0 or pH 3.0 elicited small currents (Fig. 5a). Strikingly, in oocytes expressing wild-type ASIC2a together with ASIC3-H72N_H73N similarly only pH 4.0 or pH 3.0 elicited small currents (Fig. 5a), suggesting that wild-type ASIC2a cannot efficiently rescue the channels. In contrast, in oocytes expressing wild-type ASIC3 together with ASIC2a-H72N, currents were elicited by pH 6.0 or below and at pH 4.0 peak currents had an amplitude of >1 μA (Fig. 5a), suggesting that wild-type ASIC3 efficiently rescued the channels. These results are in agreement with the idea that ASIC2a/3 heteromers containing two ASIC2a subunits are substantially acid-shifted compared with heteromers containing two ASIC3 subunits.

To confirm that this assay can reveal differences in subunit numbers, we repeated this experiment with mutant ASIC2a and ASIC1a subunits and injected cRNA ratios that should lead to the predominant formation of heteromers[10]. Because, the ASIC1a/2a heteromer has a flexible 2:1/1:2 stoichiometry[10], we expected that both ASIC1a and ASIC2a should be able to rescue the function of mutant channels. This was indeed the case: not only ASIC1a was able to rescue channels that contained ASIC2a-H72N but also ASIC2a rescued channels containing ASIC1a-H72N_H73N (Fig. 5b). This result confirms that this assay can resolve the subunit stoichiometry of ASICs, operating at physiological pH values.

**The two ASIC2a/3 heteromers can be distinguished pharmacologically**. To test whether the two ASIC2a/3 heteromers can be distinguished pharmacologically, we determined their inhibition by the non-steroidal anti-inflammatory drug diclofenac. Diclofenac inhibits the sustained current of ASIC3[34] and the peak current of ASIC2a but not of ASIC3[35]. Because the sustained current of ASIC3 and the 3-2a-3 concatemer in oocytes is small (Fig. 3, Table 3), we focused our analysis on the inhibition of the peak current at pH 4.0. Like previously reported[35], 500 μM diclofenac reduced the ASIC2a peak current to $0.68 \pm 0.06$ of control ($p = 0.0002$, paired t-test, $n = 10$); the peak current of ASIC3 was not reduced ($1.02 \pm 0.05$; $p = 0.58$, $n = 12$; Fig. 6a, b). The peak current of the 2a-3-2a concatemer was reduced ($0.76 \pm 0.1$; $p = 0.0002$, $n = 13$), was similar to ASIC2a homomers ($p = 0.40$, ANOVA followed by Tukey´s test) but different from ASIC3 homomers ($p < 0.0001$, ANOVA; Fig. 6a, b). In contrast, the peak current of the 3-2a-3 heteromer was not reduced ($0.99 \pm 0.16$; $p = 0.73$, $n = 13$), was different from ASIC2a ($p < 0.0001$, ANOVA) but similar to ASIC3 ($p = 0.91$; ANOVA; Fig. 6a, b). These results show that diclofenac can indeed differentiate between the two heteromers; the 2a-3-2a concatemer behaves like homomeric ASIC2a and the 3-2a-3 concatemer like homomeric ASIC3.

We finally asked whether the differential pharmacology of the two heteromers would allow us to identify the presence of the two heteromers in oocytes co-expressing ASIC2a and ASIC3. We injected oocytes with three different ratios of ASIC2a and ASIC3 cRNA (5:1, 1:1, and 1:5) and determined inhibition by diclofenac (Fig. 6c, d). Peak current amplitudes of oocytes injected with a 5:1 ratio (ASIC2a:ASIC3) were reduced by diclofenac to $0.76 \pm 0.26$ of control ($p = 0.062$, paired t-test), in between oocytes expressing homomeric ASIC2a ($0.61 \pm 0.11$; $p = 0.70$, ANOVA) or homomeric ASIC3 ($0.88 \pm 0.08$; $p = 0.27$). In contrast, peak current amplitudes of oocytes injected with a 1:1 ratio were not reduced by diclofenac ($0.91 \pm 0.21$ of control; $p = 0.55$) and were different from homomeric ASIC2a ($p = 0.014$) but similar to ASIC3 ($p = 0.99$). Likewise, peak current amplitudes of oocytes injected with a 1:5 ratio were not reduced by diclofenac ($1.09 \pm 0.32$ of control; $p = 0.10$), were different from homomeric ASIC2a ($p < 0.0001$) but similar to ASIC3 ($p = 0.10$; Fig. 6d). These results suggest that oocytes injected with a 5:1 ratio (ASIC2a:ASIC3) expressed a mixed population of ASICs, possibly both ASIC2a/3 heteromers together with homomeric ASIC2a, and that oocytes injected with a 1:1 or a 1:5 ratio expressed mainly the ASIC2a/3 heteromer with two ASIC3 subunits together with homomeric ASIC3.

We tested the contribution of the different subpopulations to the ASICs in oocytes co-expressing ASIC2a and ASIC3 by fitting the concentration-response relationships from Fig. 1 with the sum of four Hill functions; for two Hill functions we used $pH_{50}$ and Hill coefficients determined for homomeric ASIC2a and ASIC3 and for the other two we used the values determined for the two concatemers (2a-3-2a and 3-2a-3), respectively (Table 3). When performed for activation curves, this analysis revealed that oocytes injected with equal amounts of ASIC2a and ASIC3 expressed mainly the 3-2a-3 channel; only with higher amounts of the ASIC2a subunit (ASIC2a:ASIC3—5:1), the 2a-3-2a channel became predominant (Fig. 7a). In contrast, for SSD curves, this analysis revealed a predominant presence of the ASIC2a homomer in oocytes injected with equal amounts of ASIC2a and ASIC3 (Fig. 7b). While the 3-2a-3 channel was predominant in oocytes expressing higher amounts of the ASIC3 subunit

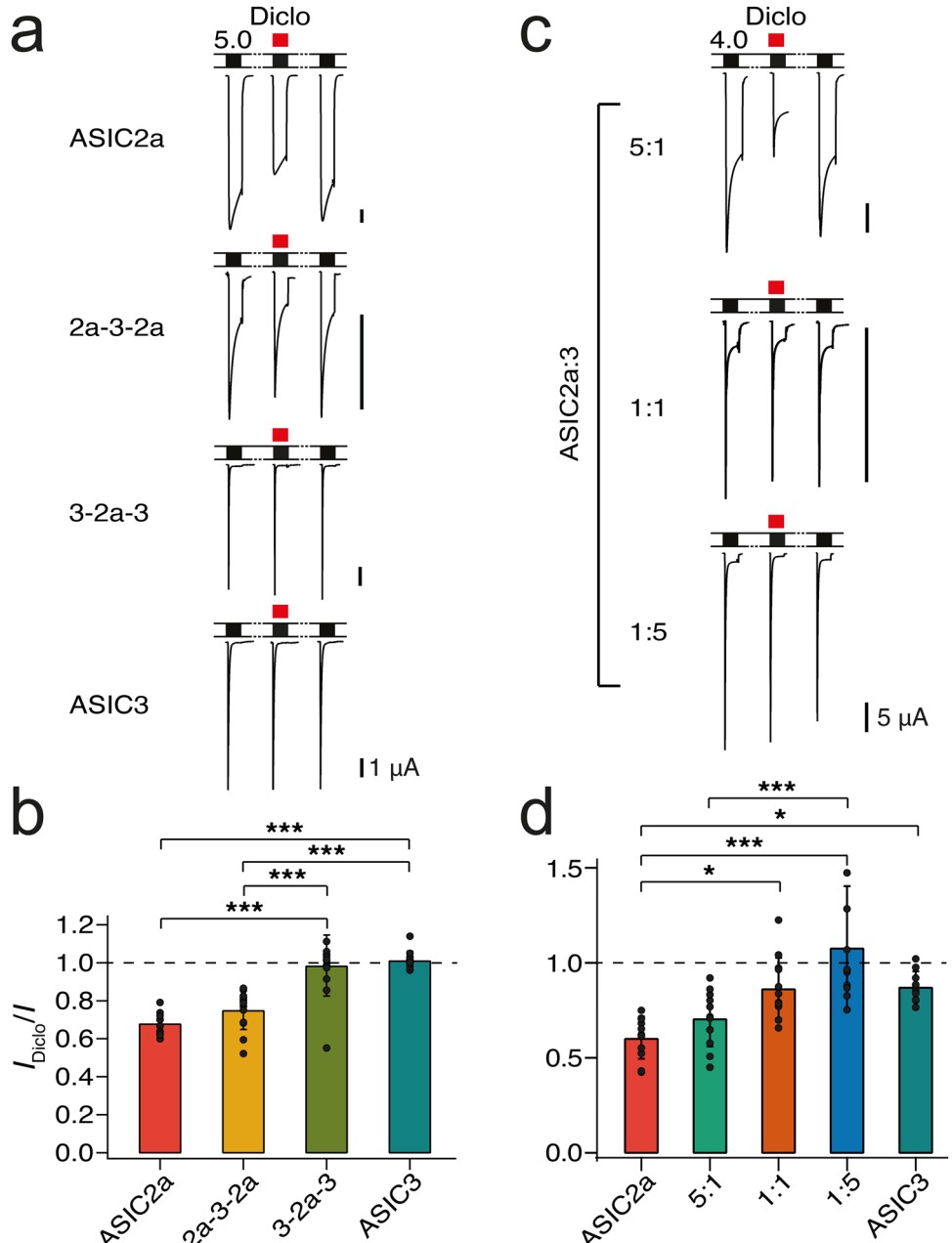

**Fig. 6 Diclofenac differentiates between the two heteromers. a** Representative current traces from measurements determining inhibition of homomeric ASIC2a and ASIC3 and ASIC concatemers by 500 μM diclofenac. ASICs were activated with pH 5.0. **b** Bar graph summarizing inhibition of ASICs by diclofenac. Peak currents in the presence of diclofenac were normalized to the peak current before application of diclofenac. The number n of independent oocytes was (from left to right): 10, 13, 13, and 12, respectively. **c** Representative current traces from measurements determining inhibition by 500 μM diclofenac for oocytes injected with different ratios of ASIC2a and ASIC3. ASICs were activated with pH 4.0. **d** Bar graph summarizing inhibition of ASICs by diclofenac. The number n of independent oocytes was (from left to right): 11, 11, 12, 12, and 10, respectively. Error bars represent SEM. *$p > 0.5$; **$p < 0.01$, ***$p < 0.001$ (ANOVA, followed by Tukey's test).

(ASIC2a:ASIC3—1:5), the 2a-3-2a made only a small contribution in all conditions (Fig. 7b). Irrespective of this inconsistency, the fit with the sum of four Hill functions did not reveal results that would have been expected for random assembly of ASIC2a and ASIC3 subunits (Fig. 7c).

## Discussion
Our results provide strong evidence that the two ASIC2a/3 heteromers have different properties, in particular proton sensitivity of activation and of SSD. While pH$_{50}$ of activation and of SSD of

concatemers containing two ASIC3 subunits and one ASIC2a subunit were intermediate between those of the two homomeric channels (ASIC2a and ASIC3), those of the concatemers containing only one ASIC3 subunit and two ASIC2a subunits resembled those of the ASIC2a homomer. Strikingly, when the two subunits were co-expressed in *Xenopus* oocytes, there was an ASIC population with properties that were intermediate between those of the two homomeric populations and that matched well the properties of the concatemers with two ASIC3 subunits, suggesting that this heteromeric population had a composition with two ASIC3 subunits. In our interpretation, the heteromeric

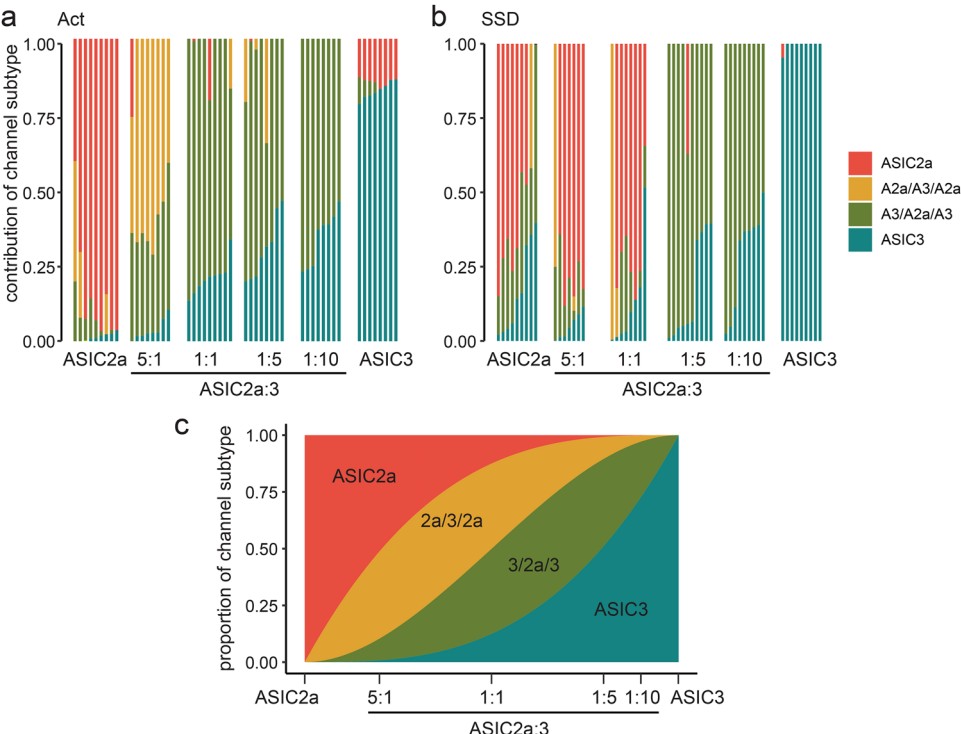

**Fig. 7 Analysis of mixed ASIC populations by using data of concatemer recordings.** pH response relationships for activation (**a**) and SSD (**b**) from Fig. 1 were reanalyzed by fitting with the sum of four Hill functions. Each Hill function had fixed $pH_{50}$ values and Hill coefficients corresponding to ASIC2a, ASIC3, the heteromer with two ASIC2a subunits and the heteromer with one ASIC2a subunit, respectively (data taken from Fig. 3). The contribution of each Hill function, corresponding to an ion channel subpopulation, is displayed as color code for individual recordings. The cRNA injection condition is indicated on the x-axis. **c** The proportions of ion channel subpopulations that would be expected from a random assembly are shown as a function of the ratio of ASIC2a and ASIC3 subunits.

population with two ASIC2a subunits could not be differentiated from the homomeric ASIC2a population.

This situation is in stark contrast to the ASIC1/2a heteromer, where the two heteromers randomly assemble[10] and have very similar functional properties rendering their differentiation difficult[25]. Thus, it appears that the contribution of individual subunits in these two heteromers (ASIC2a/3 and ASIC1a/2a) is different. Both ASIC1a and ASIC3 are subunits with high affinity for protons; ASIC2a in contrast is a subunit with low affinity for protons. It appears that for the ASIC1a/2a heteromer the first high-affinity (ASIC1a) subunit makes a large contribution to proton sensitivity, the second high-affinity subunit makes almost no further contribution, and the third high-affinity subunit makes again a larger contribution. For the ASIC2a/3 heteromer, however, it appears that a single high-affinity subunit (ASIC3) makes almost no contribution to the proton sensitivity and that two high-affinity subunits are necessary to shift proton sensitivity remarkably. The third high-affinity subunit makes again a larger contribution. Proton-sensitivity of the ASIC2a/3 concatemers with two ASIC3 subunits were indeed similar to those of the ASIC1a/2a heteromers (irrespective of their stoichiometry)[25]. Interestingly, $\tau_{des}$ of the 2a-3-2a concatemer was significantly smaller than for the 2a-2a-2a concatemer or for ASIC2a (1.9 s compared with 9.9 s and 10.8 s, respectively; Table 3). Likewise, $I_{5s}/I_{peak}$ was smaller for the 2a-3-2a concatemer than for the 2a-2a-2a concatemer or for ASIC2a (Table 3), suggesting that a single ASIC3 subunit makes a stronger contribution to desensitization than to activation.

The interpretation that the two ASIC2a/3 heteromers have different proton sensitivity is confirmed by the analysis of heteromers containing mutant subunits. Although there are different

possible interpretations of this experiment, based on the results with the ASIC1a concatemers, we draw two conclusions. First, that only channels with two wild-type subunits are functional. Second, that a single mutant subunit acid-shifts $pH_{50}$ of activation. Therefore, in the case of the ASIC1a/2a heteromer, when the ASIC1a subunit was mutated, only the heteromer with two (wild type) ASIC2a subunits was still active. This mutant ASIC1a/2a with the 1:2 stoichiometry was acid-shifted with respect to the wild-type heteromer characterized previously[25], such that pH 5.0 still elicited robust currents (Fig. 5b). In contrast, in the case of the ASIC2a/3 heteromer with the 2:1 stoichiometry, already the wild-type heteromer had a low pH sensitivity like ASIC2a (Fig. 3). Thus, a further acid-shift by the mutation of ASIC3 would render this channel largely unresponsive to protons (down to pH 3), just as we observed (Fig. 5a). In contrast, when the ASIC2a subunit was mutated, only the heteromer with two (wild type) ASIC3 subunits was still active. Because the ASIC2a/3 heteromer with the 1:2 stoichiometry had a much higher proton sensitivity than the one with the 2:1 stoichiometry, comparable to the ASIC1a/2a heteromers[25], mutation of the one ASIC2a subunit, while acid-shifting the proton sensitivity of this channel, would still leave a channel in which pH 5.0 elicited robust currents (Fig. 5a). Thus, these results are consistent with the interpretation that the ASIC2a/3 heteromer with the 2:1 stoichiometry is substantially acid-shifted compared with the ASIC2a/3 heteromer with a 1:2 stoichiometry. We note that we cannot be sure that our initial conclusions from the 1a-1a-1a concatemers (only channels with two wild-type subunits are functional; a single mutant subunit acid-shifts $pH_{50}$ of activation) also apply to ASICs containing other subunits than ASIC1a. However, the experiments with the mutation-containing ASIC1a/2a heteromers provide

independent support for the interpretation of the experiments with the mutation-containing ASIC2a/3 heteromers.

The low proton sensitivity of the ASIC2a/3 heteromer with two ASIC2a subunits, which was difficult to differentiate from the ASIC2a homomer, raises the question of whether this heteromer exists at all. Biochemical analysis revealed, however, that concatemers containing two ASIC2a subunits are stable and present at the plasma membrane (Supplementary Fig. 1). Moreover, these concatemers were functional, irrespective of the position of the single ASIC3 subunit within the trimer (Fig. 3). Therefore, we have no reason to refute the possibility that this channel exists.

Due to its low proton sensitivity, homomeric ASIC2a is usually not considered to participate in proton-sensing in the CNS or the PNS. Because ASIC2a/3 heteromers with the 2:1 stoichiometry have strongly acid-shifted $pH_{50}$ values, like ASIC2a, they probably also do not contribute to sensing acidosis in situ. In a study, in which ASICs of mouse cardiac afferents were characterized, the authors reported a $pH_{50}$ of 5.6 for cardiac afferents and for ASIC2a/3 heteromers heterologously expressed in CHO cells and they concluded that the ASIC2a/3 heteromer is the ASIC of mouse cardiac afferents. This $pH_{50}$ value is indeed in excellent agreement with the values for the ASIC2a/3 concatemers with two ASIC3 subunits ($pH_{50} = 5.3$–$5.7$), suggesting that the ASIC2a/3 heteromer of mouse cardiac afferents indeed has a 1:2 stoichiometry. However, $\tau_{des}$ at pH 4.0 was substantially slower in their study ($\tau_{des} = 1.9$ s for cardiac afferents and $\tau_{des} = 1.8$ s for ASIC2a/3)[28] than for the 3-2a-3 concatemer ($\tau_{des} = 0.6$ s), possibly due to differences in species or expression system (DRG neurons and CHO cells vs. oocytes) or both.

Do ASIC2a and ASIC3 subunits randomly assemble like ASIC1a and ASIC2a[10] or is there preferential assembly of one ASIC2a/3 heteromer? In an ideal situation, injecting equal amounts of ASIC2a and ASIC3 cRNA into oocytes would result in equal amounts of both proteins. However, when injected alone, ASIC3 had lower expression than ASIC2a (Fig. 3), suggesting that the translational efficiency or stability of ASIC3 in oocytes was lower than those of ASIC2a. This precludes clear conclusions on preferential assembly. Nevertheless, even though less ASIC3 might be present in oocytes injected with both cRNAs in a 1:1 ratio, proton sensitivity of activation, $I_{5s}/I_{peak}$, and sensitivity to diclofenac of ASICs in these oocytes was clearly more similar to the properties of the 3-2a-3 concatemer than of the 2a-3-2a concatemer (Tables 1–3). Moreover, fitting the concentration-response relationships of activation curves from these oocytes with the sum of four Hill functions revealed a clear under-representation of the 2a-3-2a channel, while the 3-2a-3 channel was predominant. On the other hand, analysis of SSD curves did not yield clear results in favor of preferential assembly of the 3-2a-3 channel. Thus, while our results do not allow clear conclusions on random or preferential assembly of ASIC2a and ASIC3 subunits, they speak in favor of a preferential assembly of the ASIC2a/3 heteromer containing two ASIC3 subunits.

What is the relative expression of the two ASIC2a/3 heteromers in the PNS? A previous study, using a semi-quantitative biochemical approach, concluded that the main functional ASICs in the brain are ASIC1a homomers and ASIC1a/2a heteromers, that the number of ASIC2a homomers is negligible, and that the ASIC1a/2a heteromers predominantly have a 2:1 stoichiometry. Moreover, in many parts of the brain, heteromers constitute ≥50% of all functional ASICs[20]. The relative quantities of ASIC2a and ASIC3 proteins in individual DRG neurons is unknown and may vary from neuron to neuron. Irrespective of these uncertainties, our results suggest that the expression of ASIC2a relative to ASIC3 needs to be more tightly tuned than the relative expression of ASIC2a and ASIC1a, to avoid the formation of the low-sensitivity ASIC2a/3 heteromer with the

2:1 stoichiometry. Preferential assembly of the ASIC2a/3 heteromer with the 1:2 stoichiometry (ASIC2a:3) would be one way to avoid this problem. Whether the heteromer with the 2:1 stoichiometry exists in DRG neurons, whether it even provides these neurons with a selective advantage to tune the sensitivity of their ASICs or whether this heteromer has a different functional role than acid-sensing are open questions that need to be determined in the future.

## Methods

**Plasmids and RNA synthesis.** For all experiments, the rat orthologs of ASIC1a, ASIC2a, or ASIC3 were used. Capped cRNA was synthesized in vitro with the mMessage mMachine kit using the SP6 RNA polymerase (Ambion, Austin, TX). All plasmids used in this study are available from the corresponding author upon reasonable request.

**Generation of concatemeric ASICs.** First, using PCR we generated constructs harboring single ASIC1a, ASIC2a or ASIC3 subunits flanked by linker sequences and restriction sites. These constructs were cloned into pBluescript II KS(-) and verified by Sanger sequencing. Next, these constructs were joined to the desired concatemer. First, the construct containing the central subunit of a desired concatemer was subcloned into the oocyte expression vector pRSSP (see Supplementary Figure 2, SacII/KpnI). The construct containing the last subunit (position 3, SpeI/XmaI) and the construct containing the first subunit (position 1, NotI/XbaI) were added by subcloning to yield a plasmid containing the complete concatemer. Success of the subcloning steps were checked by restriction digestion and agarose gel-electrophoresis. The linker sequences were as follows (protein sequences): between first and second subunit: GRSRAGSAGSAGSAGSAGS; between second and third subunit: CRTSAGSAGSAGSAGSAGS.

Some minor changes were introduced into the constructs containing ASIC2a or ASIC3 compared with ASIC1a. First, a Kozak-sequence from the Alfalfa mosaic virus was inserted prior to the initial start codon[36], and second, sequences prior to the start-codons of subunits two and three were adjusted to be as different as possible from Kozak-sequences. In constructs with polyhistidine-tags, seven histidines were inserted after the methionine.

**Handling of *Xenopus laevis* oocytes.** Animal care and surgery of frogs were conducted under protocols approved by the State Office for Nature, Environment and Consumer Protection (LANUV) of North Rhine-Westfalia (NRW), Germany, and were performed in accordance with LANUV NRW guidelines. The procedure included surgical extraction of oocytes from female *Xenopus laevis* frogs, and enzymatic treatment with collagenase type 2. Healthy stage V or VI oocytes were selected and injected with cRNA encoding ASIC subunits or concatemers, followed by incubation in Oocyte Ringer 2 (OR-2) medium at 19 °C for 24–72 h. OR-2 medium contained (in mM): 82.5 NaCl, 2.5 KCl, 1.0 $Na_2HPO_4$, 5.0 HEPES, 1.0 $MgCl_2$, 1.0 $CaCl_2$, supplemented with 0.5 g/l polyvinylpyrrolidone, 1000 U/l penicillin and 10 mg/l streptomycin; the pH was adjusted to 7.3.

If not stated otherwise, the following amounts of cRNA were injected for the respective ion channel constructs (in ng per oocyte): 0.02–0.08 for wild-type ASIC1a, 0.66–3.3 for wild-type ASIC2a, wild-type ASIC3 and ASIC2a/3 heteromers, 0.12–20 for ASIC1a concatemers, and 15–25 for ASIC2a-ASIC3 concatemers. In experiments with mutant subunits (Fig. 6), to obtain predominantly heteromeric channels ~4–10 ng of cRNA in the following ratios were injected: ASIC2a:ASIC3 = 1:5; ASIC2a(H73N):ASIC3 = 1:2.5; ASIC2a:ASIC3(H72N_H73N) = 1:10; ASIC2a(H73N):ASIC3(H72N_H73N) = 1:5. Thus, the relative amount of a mutant cDNA was doubled relative to the ratio of 1:5 for ASIC2a:ASIC3, which showed a high proportion of putative heteromers, to account for the inactivity of homomeric mutant channels. For biochemical detection of concatemers (Supplementary Fig. 1), the following amounts of cRNA were injected (in ng per oocyte) in the order of the figure from left group no. 1 (ASIC2a) to right group no. 10 (ASIC3): 64, 36, 63, 52, 76, 33, 29, 21, 32, 62.

**Electrophysiology.** For two-electrode voltage clamp, the Turbo Tec 03X amplifier (npi electronic, Tamm, Germany) was used. Experiments were performed at a holding potential of −70 mV and at room temperature (21–23 °C). The solution exchange was carried out with the help of a pump system controlled by the software Cellworks (version 5.5.1; npi electronic). Within the solution exchange system, oocytes were fixed to a glass capillary via negative pressure to enable insertion of micropipettes filled with 3 M KCl as intracellular electrodes. Proton affinity of activation of ASICs was determined by alternately applying solutions with a pH of 7.4 for 60 s and solutions with stepwise decreasing pHs for 10 s if not stated otherwise. Proton affinity of SSD was determined by alternately pre-conditioning with decreasing pHs (starting from pH 7.4) for 2 min and by applying pH 4.0 for 10 s. Measurements were taken from distinct oocytes if not stated otherwise.

**ASIC2a-3 biochemistry.** The expressed ASIC proteins were labeled, purified, resolved, and detected using previously described methods[37]. Briefly, cRNA-

injected oocytes were metabolically labeled with L-[$^{35}$S]methionine (Perkin Elmer NEG009A) overnight at 19 °C. After a subsequent one-day chase period, the oocytes were washed in OR-2, pH 8.5, and the outer surface of intact oocytes was labeled by the membrane-impermeable, amine-reactive infrared dye IR800 NHS ester (LI-COR Biosciences) for 1 h. Oocytes were washed three times in OR-2 to remove unbound IR800 dye and homogenized in 1% digitonin (w/v, Serva Electrophoresis, product no. 19551, water soluble quality) in phosphate-buffered saline, pH 8.0. Digitonin-solubilized ASIC proteins were purified via their His-tag by Ni-NTA chromatography (Qiagen) and immediately resolved by SDS-urea-PAGE on 4–10% acrylamide gradient gels. Wet and dried PAGE gels were scanned for IR800 fluorescence and $^{35}$S radioactivity using a LI-COR Odyssey scanner and a PhosphorImager (Storm 820, GE Healthcare), respectively. Unprocessed scans of the gel presented in Supplementary Fig. 1 are shown in Supplementary Fig. 3.

**Statistics and reproducibility**. We assumed normal distribution of our data but did not formally check for it. The sample sizes ($n$) indicate the number of individual oocytes. Oocytes originating from at least two different frogs were analyzed; oocytes were randomly allocated to the experimental groups. Results are reported as mean ± SEM, if not stated otherwise, but individual data points are shown on the figures, allowing to assess the variation of the data. $P$ values were determined by ANOVA followed by Tukey's test. Graphs and electrophysiological traces were plotted with Igor Pro (version 5.0.3, WaveMetrics, Lake Oswego, OR) or ggplot2.

Electrophysiological data were acquired and analyzed using Cellworks reader (version 3.7; npi electronic). Data shown in Figs. 1–3 were analyzed with a custom script in Igor Pro and subsequently manually checked.

pH-response curves were fit to the Hill equation:

$$I = \frac{I_{max} - I_0}{1 + (\frac{10^{-pH50}}{10^{-pH}})^H} \quad (1)$$

where $I_{max}$ is the maximal current, $pH_{50}$ the pH at which half-maximal current amplitude was reached, and $H$ the Hill-coefficient.

Data in Fig. 1 and Table 1 were fitted with Igor Pro. Usually, $I_{max}$ was fixed to 1. Only for activation recordings of ASIC2a homomers and ASIC2a-3 5:1 heteromers, $I_{max}$ was not fixed, because their activation curves were in a low pH range. Figures show currents that were normalized to the current at pH 5 (ASIC3 homomers) or to the current at pH 4 (all others). In one recording of the 1:10 (ASIC2a:ASIC3) condition, currents were normalized to the current at pH 4.6 as this resulted in a higher quality of fitting. For ASIC2a homomers and ASIC2a/3 5:1 heteromers, data were rescaled to a 0-1 range by dividing by $I_{max}$. Data for SSD recordings of homomers was normalized to the current at conditioning pH 7.4, whereas for heteromers, it was normalized to the maximal current for each heteromer.

Data in Figs. 2 and 3 and Table 3 were fitted with R (version 4.0.5; executed within RStudio, version 1.3.1093) using the nlsLM function from the R-package minpack.lm (version 1.2-1). To improve curve fitting, $I_{max}$, $pH_{50}$, and $H$ were constrained. $I_{max}$ was constrained to a value ≤ 2 in activation recordings or, in SSD recordings, to a value smaller than the highest normalized current of SSD recordings plus 0.1. For ion channels with 2 or 3 ASIC2a subunits we did not regularly observe plateauing of currents in activation recordings. Therefore, these recordings do not contain reliable information on the underlying (or true) maximal current that could be observed when activating with lower pH values. The constraint $I_{max}$ ≤ 2 then serves as an upper estimate for the underlying (normalized) maximal current, which might lead to estimates of $pH_{50}$ that are higher than the underlying (true) $pH_{50}$ values. $pH_{50}$ and $H$ were constrained to values ≤ 99. Recordings that yielded outliers in $pH_{50}$ or $H$ were not used in Table 3 and Fig. 3. Outliers were identified using the interquartile range (IQR) method. Data points smaller than $Q1 - 1.5 \times IQR$ or larger than $Q3 + 1.5 \times IQR$ were therefore considered outliers, where Q1 and Q3 are the first and the third quartile, respectively.

For fitting mixed ASIC populations by the sum of three Hill equations, the following equation was used:

$$current = s * \left( \frac{cA2 * HillA2(pH)}{1 + cA2 + cA3} + \frac{cA3 * HillA3(pH)}{1 + cA2 + cA3} + \frac{HillHet(pH, Hcoeff, pH50)}{1 + cA2 + cA3} \right) \quad (2)$$

For two of these Hill equations (HillA2 and HillA3) we used the mean values for $I_{max}$, $pH_{50}$, and $H$ as determined for ASIC2a and ASIC3, respectively. For the third Hill equation (HillHet), $I_{max}$ was set to 1. A scaling parameter, $s$, was multiplied with the sum of the three Hill equations to allow variations in the maximal currents between different recordings. cA2, cA3, $s$, Hcoeff, and $pH_{50}$ were obtained for individual oocytes with the nlsLM function. Relative contributions of the Hill equations were then calculated by the formula $\frac{x}{1+cA2+cA3}$ where $x$ was replaced by cA2, cA3, or 1 to get the contribution of HillA2, HillA3, or HillHet.

Mixed ASIC populations were additionally fit by the sum of four Hill functions:

$$current = s * \left( \frac{HillA3(pH)}{1 + cA2 + cH1 + cH2} + \frac{cA2 * HillA2(pH)}{1 + cA2 + cH1 + cH2} \right.$$
$$\left. + \frac{cH1 * HillH1(pH)}{1 + cA2 + cH1 + cH2} + \frac{cH2 * HillH2(pH)}{1 + cA2 + cH1 + cH2} \right) \quad (3)$$

where HillA3, HillA2, HillH1, and HillH2 are Hill functions representing the ion channel populations ASIC3, ASIC2a, the heteromer with one ASIC2a subunit and the heteromer with two ASIC2a subunits, respectively. cA2, cH1, and cH2 are parameters representing the contribution of each Hill function, and $s$ is a scaling parameter. The parameters of the Hill functions ($I_{max}$, $pH_{50}$, and $H$ of HillA3, HillA2, HillH1, and HillH2) were derived from concatemer recordings. cA2, cH1, cH2, and $s$ were obtained for individual oocytes with the nlsLM function. Relative contributions of the Hill functions were calculated by the formula

$$\frac{x}{1 + cA2 + cH1 + cH2} \quad (4)$$

where $x$ was replaced by cA2, cH1, cH2, or 1 to get the contribution of HillA2, HillH1, HillH2, or HillA3.

Expected proportions of ion channel subpopulations in the case of co-expression of two subunits and random assembly of the subunits were calculated using the following formulas, where $a$ corresponds to the percentage of expression of ASIC2a and $b$ to the percentage of expression of ASIC3:
proportion of homomeric ASIC2a: $a^3$
proportion of the ASIC2a/3 heteromer with a 2:1 stoichiometry: $3 * (a^2) * b$;
proportion of the ASIC2a/3 heteromer with a 1:2 stoichiometry: $3 * a * (b^2)$;
proportion of homomeric ASIC3: $b^3$.

The proportions of ion channel subpopulations that would be expected from a random assembly could then be calculated relative to the proportion of ASIC3 subunits.

Time constants of desensitization ($\tau_{des}$) were determined by fitting the current to the following function:

$$I = A_0 + A_1 * e^{-\frac{t}{\tau_{des}}} \quad (5)$$

where $I$ is the current at time $t$, $A_0$ the current component which did not decay, and $A_1$ the current component which did decay. Fitting was performed in R with nlsLM from the package minpack.lm.

**Reporting summary**. Further information on research design is available in the Nature Portfolio Reporting Summary linked to this article.

## Data availability

All data that support the findings of this study are included within this paper. Source data for figures can be found in Supplementary data.

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

## Acknowledgements

We thank K. Augustinowski for initial help with the characterization of ASIC1a/2a heteromers.

## Author contributions

L.F. constructed the ASIC1a concatemers and acquired and analyzed the electro-physiological data of ASIC2a/3 heteromers, ASIC1a homomers and concatemers, A.S. designed the ASIC2a/3 concatemers and A.O.B. and A.S. constructed concatemers. S.J. acquired the electrophysiological data with diclofenac. S.J. and A.S. acquired the elec-trophysiological data of ASIC2a/3 concatemers and A.S. analyzed the resulting electro-physiological data. A.D. performed the biochemical analysis of ASIC2a/3 concatemers, N.J. acquired the electrophysiological data of ASIC1a/2a heteromers, G.S. designed and analyzed the biochemical analysis of concatemers, and S.G. conceived the study; L.F., A.S., and S.G. drafted the manuscript with the help of all other authors.

## Funding

## Competing interests

The authors declare no competing interests.
