## [Peer Review File · Communications Biology]

Reviewers' comments:

Reviewer #1 (Remarks to the Author):

This manuscript by Fischer et al. investigates the stoichiometry of ion channel ASIC2a/3 heteromers. The authors found that ASIC2a/3 channels with a 1:2 stoichiometry had intermittent pH sensitivity compared to ASIC2a and -3 homomeric channels, whereas those with a 2:1 stoichiometry has properties not different from ASIC2a homomers. The experiments are well designed, the manuscript is well written, and the findings are novel and have important implications. However, some key experiments are missing, and the data does not support all the authors conclusions.

Specific comments:

1. In figure 1C it is not clear why the data points do not line up horizontally with the hashmarks on the Y-axis.
2. For pH dose responses of activation and steady-state desensitization, it is not clear how the data is normalized.
3. Can the authors speculate as to why WT ASIC3 and concatemeric ASIC3 were expressed in the plasma membrane fraction (Fig. 3), and yet their expressed produced robust currents in oocytes?
4. Page 9: please clarify "values for the 2a-2a-2a concatemer were variable between the two weeks of recording".
5. The basis of the authors conclusions rests solely on their measurements of pH sensitivity and current amplitude. On the other hand, many other properties could have been measured including but not limited to time constants of desensitization, recovery from desensitization, amplitude of sustained current, and pharmacological modulation. From the current traces in Figures 1 it appears that channels formed by ASIC2a and -3 coexpression have desensitization kinetics that are like ASIC2a homomers and are slower than ASIC3 homomers. However, the concatemeric currents shown in Figure 4 suggest that the 2a-3-2a concatemer has kinetics similar to channels formed by ASIC2a and -3 coexpression, whereas the 3-2a-3 concatemer appears to have desensitization kinetics that are much faster (resembling ASIC3 homomers). The authors should report the desensitization kinetics of all currents and discuss why the 3-2a-3 concatemer properties do not match that of channels formed by ASIC2a and -3 coexpression.
6. Some of the figure font sizes (particularly figure 4) need to be larger.
7. The conclusions of the mutant channel experiments are based on results of His residue mutations introduced into 1a-1a-1a concatemers. From these results the authors conclude that channels that include 2 mutant subunits are nonfunctional, and they assume that this is also true of channels formed by ASIC2a and -3. Since the subunits have very different properties, I don't think the authors can make this assumption without studying the effect of these mutation on both 2a-2a-2a and 3-3-3 channels.

Reviewer #2 (Remarks to the Author):

This paper by Fischer et al questions the occurrence and function of distinct heteromeric acid-sensing ion channels (ASICs). It contains a lot of good experiments that suggest that ASIC2a/ASIC3 heteromers of two different sorts can form: 2a/2a/3 and 2a/3/3. The results show that these two heteromers have different functional properties, which makes them very different from previously characterized ASIC1a/ASIC2a heteromers.

The results are clearly laid out and experiments are repeatable, the paper is well-written, the numbers and stats are good, and I think it deserves publication in this journal. However, I think it needs some

small revisions before publishing. These results are the first on this question of 2a/3 stoichiometry, and the abundance or preference of the two different (stoichiometries of) ASIC2a/ASIC3 heteromers should be analyzed/discussed. Along this line, I don't see how the experiments presented Figure 1 lead to the current interpretations. The results also offer novel insight into the contribution of different subunits to channel activation, as stoichiometry seems to affect proton sensitivity for activation, proton sensitivity for desensitization, and speed of entry into desensitization differently. This is a big question and can't be fully answered here, but it deserves some discussion in light of the interesting results. These and other smaller points are detailed below.

Figure 1

The results showing activation of currents in 2a, 3, and 2a:3 5:1 in panel A seem to match the plots for activation in panel B. But the currents in 2a:3 1:1 and 2a:3 1:5 in panel A do not seem to match the plots in panel B.

In panel C, it's not immediately clear which rows the two top activation data groups refer to – can these be moved up or down accordingly?

Line 101/102, "Variable values for pH50...of SSD"

To me it looks like most of the heteromer combinations have less variability in SSD pH50 values than the homomers! (Which is not inconsistent with your main findings.)

Lines 124-136, Figure 2, Table2

Were other alternatives considered, such as four components instead of three. I am curious, because looking at the activation plots in Figure 2A, I wonder if e.g. 2a:3 1:5 has two S curves in the range of 6.0 and 4.0. These are presumably not homomers (based on pH50 values), is it possible that there are 2a/3/3 heteromers with a pH 50 of ~5.9 and 2a/2a/3 heteromers with a pH50 of ~4.5? Or is the 3-component fit indeed better, and the second S curve in the range I mention just the 2a homomers starting to be activated between pH4.5 and 4.0? If the former is the case, you might still see a greater contribution of one of the components and this could mean a preferred 2a/3/3 or 2a/2a/3 stoichiometry. It might also reasonably fit with the 2a/2a/3 concatemer activation pH50 on lines 213-216, etc.

Why does the plot in Figure 2A not go to pH 3.5 or so, where the curves are clearly fit to?

Discussion, first paragraph – I don't see that oocytes injected with 2a and 3 are preferring to form 2a/3/3 heteromers. Isn't it possible that they form both 2a/2a/3 and 2a/3/3 heteromers, but in the pH range tested, we of course see bigger currents in the pH range of the 2a/3/3 heteromers?

Line 380, "assumptions". Should this be "assumptions" or "interpretations"? Also, I'm not sure we can extrapolate ASIC1a mutant/concatemer results to ASIC2a/3 heteromers: you've already shown that stoichiometry differently affects ASIC1a- and ASIC3-containing channels. Presumably, the effects of the ASIC1a HN mutation are conveyed from subunit-to-subunit differently in ASIC1a homomers and ASIC1a/2 heteromers, and the effects of the ASIC2a HN mutation could be conveyed from subunit-to-subunit differently again. I think the nicest controls instead come from the control experiments on page 6 tieh ASIC1a/ASIC2a heteromers, which clearly indicate the specificity of the ASIC2a/3 heteromer results.

Line 409-411, random assembly or preferential stoichiometry. Doesn't this deserve some discussion with reference to your results? By performing experiments with co-injection (allowing for random assembly) and other experiments with concatemers (forced stoichiometry) aren't you in a position to assess this? (See also my comment on contributions from different components in the multi-component fits, above.) This would also be important for physiology: if the native heteromer is

2a/2a/3, then it doesn't matter that the 2a/3/3 heteromer has greater pH sensitivity. (Although I doubt that is the case!)

It is very interesting to see that pH50s for activation and SSD, and the time constant for entry into desensitization, are differentially affected by stoichiometry. This is briefly mentioned in the results, but I think it deserves more discussion, especially regarding how little is known about contributions of the three subunits in a trimer to activation.

Line 429 – Figure X?

Line 468-473. Please refer to specific figures/panels.

Line 478-481. I don't see any more technical details in Joeres et al., 2016. Please specify (or delete). Does this refer to the oocyte/electrode/perfusion set-up?

Line 488-489 – Do these three references to your previous work really help explain the methods here?

Line 510-533 – Perhaps you could use more sub- or super-script and match the main text terms with those in the equations better. This section is difficult to follow.

Reviewer #3 (Remarks to the Author):

In this study, the authors characterized the stoichiometry of the ASIC2a/3 heteromers by combining molecular, biochemical, and electrophysiological approaches. In contrast to the ASIC1a/2a heteromers, striking functional differences were detected between the ASIC2a/3 heteromers with a 1:2 and a 2:1 stoichiometry. Specifically, while pH50 of activation and steady-state desensitization (SSD) for concatemers containing two ASIC3 subunits and one ASIC2a subunit were in-between those of the two homomeric channels (ASIC2a and ASIC3), those of the concatemers containing only one ASIC3 subunit and two ASIC2a subunits displayed characteristics that resembled homomeric ASIC2. Therefore, only ASIC2a/3 heteromers with a 1:2 stoichiometry had a proton-sensitivity intermediate between ASIC3 and ASIC2a. In contrast, the proton sensitivity of ASIC2a/3 heteromers with a 2:1 stoichiometry was strongly acid-shifted, suggesting that they are not physiologically relevant. Overall, the paper is reasonably well-written, easy to read and well-presented. The followings are some comments:

(1) All the present data were obtained on *Xenopus laevis* oocytes. One pressing question is the stoichiometry of ASIC2/3 heteromers in neurons. It would also be necessary to check whether the current results can be replicated under in vivo situations, such as the assembly and stoichiometry of ASIC2a and ASIC3 subunits in DRG neurons.

(2) Lines 62-64: Toxins and other modulators of ASICs bind at subunit interfaces (Baconguis and Gouaux 2012; Dawson and Xia 2012), rendering the knowledge of subunit stoichiometry of interest for drug development". Then do ASIC2a/3 heteromers with different stoichiometry exhibit distinct sensitivity to pharmacological agents (such as potentiation by arachidonic acid or inhibition by NSAIDs)?

(3) Lines 173-175: "In contrast to ASIC2a, ASIC3 (60 kDa protein core plus ~5 kDa provided by two N-glycans at 176NFT and 400NRS) was only visible in the 35S labeled form, but not at the plasma membrane (lane 18)". I am not clear but why is ASIC3 invisible in plasma membrane-bound form?

(4) Lines 252-254: "Thus, in contrast to pH50 values, tdes allowed to differentiate the concatemer containing one ASIC3 subunit from homomeric ASIC2a and the 2a-2a-2a concatemer". Please explain the reasons for this difference between pH50 and tdes. Why pH50 cannot differentiate 2a-3-2a with ASIC 2a-2a-2a but tdes can?

(5) What are the possible differential roles of ASIC2a/3 heteromers with different stoichiometry (eg. 2a-3-2a vs. 3-2a-3) in terms of cellular signaling or behavioral functions?

We thank the three reviewers for taking the time to critically read our manuscript and for their thoughtful suggestions to improve it. Based on their comments we made multiple changes in the text, in particular we pharmacologically characterized the different channels with diclofenac, added a new four component fit, discuss more aspects of our work, and corrected mistakes mentioned by the reviewers. In addition, we corrected minor errors in values or samples sizes (in particular in table 2), which we noted during the revision.

Below we answer all the comments of the reviewers.

Reviewers' comments:

Reviewer #1 (Remarks to the Author):

This manuscript by Fischer et al. investigates the stoichiometry of ion channel ASIC2a/3 heteromers. The authors found that ASIC2a/3 channels with a 1:2 stoichiometry had intermittent pH sensitivity compared to ASIC2a and -3 homomeric channels, whereas those with a 2:1 stoichiometry has properties not different from ASIC2a homomers. The experiments are well designed, the manuscript is well written, and the findings are novel and have important implications. However, some key experiments are missing, and the data does not support all the authors conclusions.

Thank you for your comments.

Specific comments:

1. In figure 1C it is not clear why the data points do not line up horizontally with the hashmarks on the Y-axis.

We now align horizontally the data for activation and steady-state desensitization and hope that this makes the figure more easy to understand.

2. For pH dose responses of activation and steady-state desensitization, it is not clear how the data is normalized.

We now describe in more detail how the data was normalized (lines 616-624).

3. Can the authors speculate as to why WT ASIC3 and concatemeric ASIC3 were expressed in the plasma membrane fraction (Fig. 3), and yet their expressed produced robust currents in oocytes?

In general, current measurement is more sensitive to detect ion channels than biochemistry. Current amplitudes of WT ASIC3 and concatemeric ASIC3 had a tendency to be lower than those of ASIC2a (Table 1). Moreover, overall expression of ASIC2a was much higher than that of ASIC3 (see the strong monomeric form of ASIC2a in Fig. 3 although we injected less cRNA), and we speculate that, therefore, ASIC2a-containing constructs were better synthesized and could be detected also in the plasma membrane fraction. We now added a comment in the respective section (line 170).

4. Page 9: please clarify “values for the 2a-2a—2a concatemer were variable between the two weeks of recording”.

We deleted this half-sentence. What we meant was that at very low pH (e.g. pH 3.5) in some batches of oocytes current measurements were less reliable, making the estimation of EC₅₀ values less accurate. This was already described in the sentence following the sentence the referee is referring to.

5. The basis of the authors conclusions rests solely on their measurements of pH sensitivity and current amplitude. On the other hand, many other properties could have been measured including but not limited to time constants of desensitization, recovery from desensitization, amplitude of sustained current, and pharmacological modulation. From the current traces in Figures 1 it appears that channels formed by ASIC2a and -3 coexpression have desensitization kinetics that are like ASIC2a homomers and are slower than ASIC3 homomers. However, the concatemeric currents shown in Figure 4 suggest that the 2a-3-2a concatemer has kinetics similar to channels formed by ASIC2a and -3 coexpression, whereas the 3-2a-3 concatemer appears to have desensitization kinetics that are much faster (resembling ASIC3 homomers). The authors should report the desensitization kinetics of all currents and discuss why the 3-2a-3 concatemer properties do not match that of channels formed by ASIC2a and -3 coexpression.

The problem with the desensitization kinetics is that due to the relatively slow solution exchange in measurements with *Xenopus* oocytes, desensitization kinetics is somewhat variable between batches of oocytes and different set-ups (results for channels formed by ASIC2a and ASIC3 coexpression were obtained with a similar but different set-up than results for concatemers). pH₅₀ values can be recorded with more precision and can be better reproduced (except for very acid-shifted values), explaining why we focussed our analysis on pH₅₀ values. Nevertheless, in table 1, we now also report and briefly discuss desensitization kinetics of channels formed by ASIC2a and ASIC3 coexpression.

In addition to reporting desensitization kinetics, we also analyzed pharmacological modulation by diclofenac of the two different concatamers, of the two homomers, and of oocytes expressing both subunits at different ratios (new Figure 7). These result show that diclofenac can indeed differentiate between the two heteromers.

6. Some of the figure font sizes (particularly figure 4) need to be larger.

We increased the font sizes of figures 1, 2, and 4.

7. The conclusions of the mutant channel experiments are based on results of His residue mutations introduced into 1a-1a-1a concatemers. From these results the authors conclude that channels that include 2 mutant subunits are nonfunctional, and they assume that this is also true of channels formed by ASIC2a and -3. Since the subunits have very different properties, I don't think the authors can make this assumption without studying the effect of these mutation on both 2a-2a-2a and 3-3-3 channels.

In principle, we agree with the reviewer. We would like to underline, however, that current amplitudes of ASIC1a concatemers with two mutations were much lower than for WT concatemers, yet ASIC1a expresses much better than either ASIC2a or ASIC3. We, therefore, expect that similar ASIC2a and ASIC3 concatemers would not yield meaningful results and would, therefore, not justify the considerable additional effort to generate and characterize them.

We now mention that our conclusions based on His residue mutations in 1a-1a-1a concatemers may not apply to other ASICs (lines 469-474).

Reviewer #2 (Remarks to the Author):

This paper by Fischer et al questions the occurrence and function of distinct heteromeric acid-sensing ion channels (ASICs). It contains a lot of good experiments that suggest that ASIC2a/ASIC3 heteromers of two different sorts can form: 2a/2a/3 and 2a/3/3. The results show that these two heteromers have different functional properties, which makes them very different from previously characterized ASIC1a/ASIC2a heteromers.

The results are clearly laid out and experiments are repeatable, the paper is well-written, the numbers and stats are good, and I think it deserves publication in this journal. However, I think it needs some small revisions before publishing. These results are the first on this question of 2a/3 stoichiometry, and the abundance or preference of the two different (stoichiometries of) ASIC2a/ASIC3 heteromers should be analyzed/discussed. Along this line, I don't see how the experiments presented Figure 1 lead to the current interpretations. The results also offer novel insight into the contribution of different subunits to channel activation, as stoichiometry seems to affect proton sensitivity for activation, proton sensitivity for desensitization, and speed of entry into desensitization differently. This is a big question and can't be fully answered here, but it deserves some discussion in light of the interesting results. These and other smaller points are detailed below.

Thank you for your positive comments.

Figure 1

The results showing activation of currents in 2a, 3, and 2a:3 5:1 in panel A seem to match the plots for activation in panel B. But the currents in 2a:3 1:1 and 2a:3 1:5 in panel A do not seem to match the plots in panel B.

Thank you for your comment. The currents for 2a:3 1:1 and 2a:3 1:5 in panel A were wrong. We exchanged them for current traces that are representative. Moreover, we only show traces for the pH values that were analysed in panel B.

In panel C, it's not immediately clear which rows the two top activation data groups refer to – can these be moved up or down accordingly?

Referee #1 made a similar comment. We now align horizontally the data for activation and steady-state desensitization and hope that this makes the figure more easy to understand.

Line 101/102, “Variable values for pH50...of SSD”

To me it looks like most of the heteromer combinations have less variability in SSD pH50 values than the homomers! (Which is not inconsistent with your main findings.)

We rephrased this sentence to make its meaning clearer.

Lines 124-136, Figure 2, Table2

Were other alternatives considered, such as four components instead of three. I am curious, because looking at the activation plots in Figure 2A, I wonder if e.g. 2a:3 1:5 has two S curves in the range of 6.0 and 4.0. These are presumably not homomers (based on pH50 values), is it

possible that there are 2a/3/3 heteromers with a p_H 50 of ~5.9 and 2a/2a/3 heteromers with a p_H50 of ~4.5? Or is the 3-component fit indeed better, and the second S curve in the range I mention just the 2a homomers starting to be activated between pH4.5 and 4.0? If the former is the case, you might still see a greater contribution of one of the components and this could mean a preferred 2a/3/3 or 2a/2a/3 stoichiometry. It might also reasonably fit with the 2a/2a/3 concatemer activation p_H50 on lines 213-216, etc.

We now performed also a fit with four Hill functions. These results suggest a preferred 2a-3-2a stoichiometry and are reported in lines 395-408 and new Fig. 8.

Why does the plot in Figure 2A not go to pH 3.5 or so, where the curves are clearly fit to?

We thank the reviewer for making us aware of this problem. We now modified Figure 2 and show a wider pH-range.

Discussion, first paragraph – I don't see that oocytes injected with 2a and 3 are preferring to form 2a/3/3 heteromers. Isn't it possible that they form both 2a/2a/3 and 2a/3/3 heteromers, but in the pH range tested, we of course see bigger currents in the pH range of the 2a/3/3 heteromers?

Thank you for your comment. We agree and rephrased the respective sentence (lines 428-433).

Line 380, "assumptions". Should this be "assumptions" or "interpretations"? Also, I'm not sure we can extrapolate ASIC1a mutant/concatemer results to ASIC2a/3 heteromers: you've already shown that stoichiometry differently affects ASIC1a- and ASIC3-containing channels. Presumably, the effects of the ASIC1a HN mutation are conveyed from subunit-to-subunit differently in ASIC1a homomers and ASIC1a/2 heteromers, and the effects of the ASIC2a HN mutation could be conveyed from subunit-to-subunit differently again. I think the nicest controls instead come from the control experiments on page 16 with ASIC1a/ASIC2a heteromers, which clearly indicate the specificity of the ASIC2a/3 heteromer results.

This comment is related to a similar comment of reviewer #1. We expect that ASIC2a and ASIC3 concatemers carrying HN mutations would not express well enough to yield meaningful results and would, therefore, not justify the considerable additional effort to generate and characterize them.

We now changed "assumptions" to "conclusions" (line 454) and mention that these conclusions may not apply to all ASICs (lines 469-474).

Line 409-411, random assembly or preferential stoichiometry. Doesn't this deserve some discussion with reference to your results? By performing experiments with co-injection (allowing for random assembly) and other experiments with concatemers (forced stoichiometry) aren't you in a position to assess this? (See also my comment on contributions from different components in the multi-component fits, above.) This would also be important for physiology: if the native heteromer is 2a/2a/3, then it doesn't matter that the 2a/3/3 heteromer has greater pH sensitivity. (Although I doubt that is the case!)

While we find this question very interesting, we are afraid that our experiments do not allow a clear conclusion regarding random assembly or preferential stoichiometry. The main reason is that ASIC3 seems to express less well than ASIC2a (see Figure 3). Thus, even if we inject

defined ratios of cRNA, we cannot be sure whether this translates into the same ratio of proteins (actually, figure 3 suggests this is not the case). Nevertheless, we performed a four component fit (fig. 8) and discuss the issue of random assembly or preferential stoichiometry in more detail (lines 493-507). Overall, our results suggest that the 1:2 stoichiometry (ASIC2a:ASIC3) might be preferred.

It is very interesting to see that pH50s for activation and SSD, and the time constant for entry into desensitization, are differentially affected by stoichiometry. This is briefly mentioned in the results, but I think it deserves more discussion, especially regarding how little is known about contributions of the three subunits in a trimer to activation.

We added a sentence on this topic to the Discussion (lines 446-450).

Line 429 – Figure X?

Figure 7 was meant. Error fixed.

Line 468-473. Please refer to specific figures/panels.

Done

Line 478-481. I don't see any more technical details in Joeres et al., 2016. Please specify (or delete). Does this refer to the oocyte/electrode/perfusion set-up?

We deleted the respective sentence.

Line 488-489 – Do these three references to your previous work really help explain the methods here?

We reduced the number of references to 1.

Line 510-533 – Perhaps you could use more sub- or super-script and match the main text terms with those in the equations better. This section is difficult to follow.

We rephrased this section to make it easier to follow.

Reviewer #3 (Remarks to the Author):

In this study, the authors characterized the stoichiometry of the ASIC2a/3 heteromers by combining molecular, biochemical, and electrophysiological approaches. In contrast to the ASIC1a/2a heteromers, striking functional differences were detected between the ASIC2a/3 heteromers with a 1:2 and a 2:1 stoichiometry. Specifically, while pH50 of activation and steady-state desensitization (SSD) for concatemers containing two ASIC3 subunits and one ASIC2a subunit were in-between those of the two homomeric channels (ASIC2a and ASIC3), those of the concatemers containing only one ASIC3 subunit and two ASIC2a subunits displayed characteristics that resembled homomeric ASIC2. Therefore, only ASIC2a/3 heteromers with a 1:2 stoichiometry had a proton-sensitivity intermediate between ASIC3 and ASIC2a. In contrast, the proton sensitivity of ASIC2a/3 heteromers with a 2:1 stoichiometry

was strongly acid-shifted, suggesting that they are not physiologically relevant. Overall, the paper is reasonably well-written, easy to read and well-presented. The followings are some comments:

Thank you for your positive comments.

(1) All the present data were obtained on *Xenopus laevis* oocytes. One pressing question is the stoichiometry of ASIC2/3 heteromers in neurons. It would also be necessary to check whether the current results can be replicated under in vivo situations, such as the assembly and stoichiometry of ASIC2a and ASIC3 subunits in DRG neurons.

Functional analysis of ASICs in DRG neurons is beyond the scope of the current manuscript, in particular considering that DRG neurons and their ASICs are heterogenous. But we now discuss the published properties of the presumed 2a/3 heteromer in DRGs and how they compare to our data (lines 481-492).

(2) Lines 62-64: Toxins and other modulators of ASICs bind at subunit interfaces (Baconguis and Gouaux 2012; Dawson and Xia 2012), rendering the knowledge of subunit stoichiometry of interest for drug development”. Then do ASIC2a/3 heteromers with different stoichiometry exhibit distinct sensitivity to pharmacological agents (such as potentiation by arachidonic acid or inhibition by NSAIDs)?

We thank the reviewer for his comment. According to the literature, arachidonic acid potentiates both ASIC2a and ASIC3, rendering it less likely as a tool to differentiate between the two heteromers. Moreover, in oocytes it does not potentiate neither ASIC2a nor ASIC3 (our preliminary observations). Therefore, we assessed inhibition of ASIC2, ASIC3 and the two ASIC2a/3 heteromers with different stoichiometry by the NSAID diclofenac. The results are reported in new Figure 7 and show that diclofenac indeed allows to differentiate between the two heteromers.

(3) Lines 173-175: “In contrast to ASIC2a, ASIC3 (60 kDa protein core plus ~5 kDa provided by two N-glycans at 176NFT and 400NRS) was only visible in the 35S labeled form, but not at the plasma membrane (lane 18)”. I am not clear but why is ASIC3 invisible in plasma membrane-bound form?

This comment is related to a comment of reviewer #1. Total expression of ASIC2a was much higher than of ASIC3 (see the strong monomeric form of ASIC2a in Fig. 3), and we speculate that, therefore, ASIC2a-containing constructs were better synthesized and could be detected also in the plasma membrane fraction. We now added a comment in the respective section (line 170).

(4) Lines 252-254: “Thus, in contrast to pH50 values, tdes allowed to differentiate the concatemer containing one ASIC3 subunit from homomeric ASIC2a and the 2a-2a-2a concatemer”. Please explain the reasons for this difference between pH50 and tdes. Why pH50 cannot differentiate 2a-3-2a with ASIC 2a-2a-2a but tdes can?

This comment is related to a comment of reviewer #1. We added a sentence on this topic to the Discussion (lines 446-450).

(5) What are the possible differential roles of ASIC2a/3 heteromers with different stoichiometry (eg. 2a-3-21 vs. 3-2a-3) in terms of cellular signaling or behavioral functions?

While we find this question very interesting, we are afraid that answering it would be quite speculative at this stage. We feel that just guessing or raising speculations, beyond what we already wrote at the very end of the Discussion, at this point will not serve the readers.

REVIEWERS' COMMENTS:

Reviewer #1 (Remarks to the Author):

The authors have adequately addressed the concerns of this reviewer.

Reviewer #2 (Remarks to the Author):

All of my issues have been addressed. I think the manuscript is now much clearer and the conclusions a better match with the existing and the new data.

Reviewer #3 (Remarks to the Author):

The resubmission from Fischer et al. includes additional experiments and analyses which significantly strengthen the manuscript and satisfy the reviewer's initial concerns. The authors have also nicely addressed the comments of other reviewers. I have no further comments for this manuscript.